# Horse Y chromosome assembly displays unique evolutionary features and putative stallion fertility genes

Jan E. Janečka[1], Brian W. Davis[2], Sharmila Ghosh[2], Nandina Paria[3], Pranab J. Das[4], Ludovic Orlando[5,6], Mikkel Schubert[5], Martin K. Nielsen[7], Tom A.E. Stout[8], Wesley Brashear[2], Gang Li[2], Charles D. Johnson[9], Richard P. Metz[9], Al Muatasim Al Zadjali[1], Charles C. Love[2], Dickson D. Varner[2], Daniel W. Bellott[10], William J. Murphy[2], Bhanu P. Chowdhary[2,11] & Terje Raudsepp[2]

Dynamic evolutionary processes and complex structure make the Y chromosome among the most diverse and least understood regions in mammalian genomes. Here, we present an annotated assembly of the male specific region of the horse Y chromosome (eMSY), representing the first comprehensive Y assembly in odd-toed ungulates. The eMSY comprises single-copy, equine specific multi-copy, PAR transposed, and novel ampliconic sequence classes. The eMSY gene density approaches that of autosomes with the highest number of retained X–Y gametologs recorded in eutherians, in addition to novel Y-born and transposed genes. Horse, donkey and mule testis RNAseq reveals several candidate genes for stallion fertility. A novel testis-expressed XY ampliconic sequence class, *ETSTY7*, is shared with the parasite *Parascaris* genome, providing evidence for eukaryotic horizontal transfer and inter-chromosomal mobility. Our study highlights the dynamic nature of the Y and provides a reference sequence for improved understanding of equine male development and fertility.

[1] Duquesne University, Pittsburgh, PA 15282, USA. [2] Texas A&M University, College Station, TX 77843, USA. [3] Texas Scottish Rite Hospital for Children, Dallas, TX 75219, USA. [4] ICAR-National Research Centre on Pig, Guwahati, Assam 781131, India. [5] Natural History Museum of Denmark, 1350K Copenhagen, Denmark. [6] Université de Toulouse, Université Paul Sabatier, 31000 Toulouse, France. [7] University of Kentucky, Lexington, KY 40546-0099, USA. [8] Utrecht University, 3512 JE Utrecht, Netherlands. [9] Texas A&M AgriLife Research, College Station, TX 77843, USA. [10] Whitehead Institute in Cambridge, Cambridge, MA 02142, USA. [11] United Arab Emirates University, Al Ain 15551, UAE. Correspondence and requests for materials should be addressed to B.P.C. (email: bchowdhary@uaeu.ac.ae, email: bpcajm@gmail.com) or to T.R. (email: traudsepp@cvm.tamu.edu)

The eutherian sex chromosomes evolved from a pair of autosomes that diverged around 180 million years ago (MYA) after the Y chromosome acquired a male determining locus[1]. The majority of the Y chromosome decayed in size and gene content as it gradually lost the ability to recombine with the X chromosome through several inversions. Only the short pseudoautosomal region (PAR) maintained sequence similarity and pairing during meiosis[2,3]. Thus the main portion of the Y chromosome is male specific (MSY), haploid, and does not participate in crossing over. These features have led to the accumulation of male-benefit genes and the expansion of extensive ampliconic regions[4–6].

The Y chromosome is the most rapidly evolving nuclear chromosome studied thus far[6]. Its evolutionary dynamics and structural complexity, including the acquisition, loss and amplification of genes and DNA sequences, varies across mammals, even for closely related species such as the human and chimpanzee[6–8]. Despite its importance for understanding male biology, particularly development, and spermatogenesis, the number of sequenced eutherian Y chromosomes is low with only four completed (human[5], chimpanzee[6], rhesus macaque[7,9] and mouse[10]) and eight partial (gorilla[11], marmoset[12], rat[12], dog[13], cat[13], pig[14], bull[12], and opossum[12]). This is partly a consequence of the repetitive nature, lineage-specific content, and poor representation of the haploid Y in most genome assemblies, which in turn greatly limits our understanding of this chromosome.

Horses are an economically and culturally important domestic species. Because stallions are typically selected based on pedigrees and athletic performance, one of the most common concerns is stallion subfertility. Given the large Y-linked contribution to infertility in men[15,16], it is expected that similar important regulators of male biology are also present in the horse Y. The horse Y chromosome remains, however, poorly characterized, as the horse reference genome is from a female[17].The gene content of the horse Y has been examined from sequencing of cDNA libraries, and includes several testis-specific transcripts unique to the equid lineage[18]. The horse Y chromosome is also found to be depauperate in variation, with most common haplotypes coalescing within the last millennium[19].

Herein we generate the first comprehensive assembly and functional annotation of the male-specific region of the horse Y (eMSY) to elucidate the evolution and function of the chromosome. This is the first representative Y assembly for odd-toed ungulates (order Perissodactyla). We discuss novel features of Y chromosome evolution and their ramifications for stallion biology including the retention of the most X–Y paralogs of any species studied thus far, a recent PAR transposition potentially related to XY sex reversal, amplification of testes-expressed genes, and identification of a novel sequence class which shows strong evidence for horizontal transfer with an intestinal parasite *Parascaris*. Our assembly fills a major gap in the horse reference genome and provides novel insights into the evolution and function of this unique chromosome.

## Results and discussion
The horse Y chromosome is cytogenetically comparable in size to the smallest equine autosomes, around 40–50 million base-pairs (Mbp)[17,20]. According to chromosome banding patterns, two-thirds of the Y is heterochromatic, with a small distal euchromatic segment containing eMSY and the PAR (~2 Mbp) [21] (Fig. 1a). To sequence the eMSY, we first mapped a tiling path of 192 bacterial artificial chromosome (BAC) clones (Supplementary Note 1 and Supplementary Fig. 1) encompassing the eMSY from the pseudoautosomal boundary (PAB) to Y heterochromatin, leaving approximately 2–3 Mbp in

unmapped gaps (Fig. 1b). The tiling path was supported by 265 linearly ordered sequence tagged sites (STSs) (Supplementary Data 4) and fluorescence in situ hybridization (FISH) experiments, which confirmed clone overlaps and helped orient the contigs, estimate the size of gaps and clone copy numbers (Supplementary Figs. 2–3). We sequenced 94 tiling-path BACs with the highest redundancy in the multi-copy region (Supplementary Fig. 1 and Supplementary Data 5). The final eMSY de novo assembly was 9,497,449 bp (Fig. 1b and Supplementary Table 1), however we estimate the size of eMSY to be approximately 12 Mbp when unmapped gaps are included.

**Unique features of the horse Y chromosome.** Over half (54%) of eMSY was composed of various interspersed repeats (Supplementary Table 2 and Supplementary Note 3) with L1 LINE elements (34%) being predominant. In the remaining non-repetitive eMSY, we identified four distinct sequence classes: single-copy (~60%), multi-copy/ampliconic (~37%), PAR transposed, and novel XY ampliconic array (Fig. 1b). Single-copy sequences contained the majority of ancestral X–Y (25 of 29) and autosomal transposed (7 of 10) genes. Nearly all multi-copy/ampliconic sequences localized between eMSY:1,000,000–4,700,000 bp. Characteristic to these regions was the presence of high-identity repeats (Fig. 1c and Supplementary Fig. 4). Within these, we observed regions which contained both intact copies of genes and those with an incomplete numbers of exons. The third sequence class represented a 125,171 bp transposition (Y:429,056–554,227) from the PAR (X:495,796–603,635) containing the *ARSFY* and *ARSHY* genes in inverted orientation (Fig. 1b). The average sequence identity between the PAR and eMSY in this region was 98.8%, suggesting that the transposition is of recent origin. FISH with eMSY BAC clones confirmed that this transposition is shared by other equids, including the donkey (*Equus asinus*; EAS), the quagga plains zebra (*E. burchelli*; EBU), and the Hartmann's mountain zebra (*E. zebra hartmannae*; EZH) (Fig. 2). The occurrence of the transposition in both caballine (the horse) and non-caballine species (asses and zebras) suggests that it was already present in the most recent common ancestor of both clades some 4.0–4.5 MYA[22]. The high sequence similarity with the X-PAR likely facilitates ectopic recombination between the eMSY and the PAR, providing an explanation for the massive deletions in the eMSY that are occasionally found in horses with disorders of sexual development[23]. Finally, the most proximal 250 kb of eMSY harbored a novel ampliconic sequence class with arrays of equine testis-specific transcript 7 (*ETSTY7*). Such arrays are characteristic to all Y chromosomes studied so far and thought to be needed to stabilize Y gene content and protect spermatogenesis genes[1,12]. However, ampliconic arrays are not conserved and show unique, species-specific features of origin, distribution, and sequence properties. Analysis of a short PCR product of this region by FISH revealed that *ETSTY7* arrays spread throughout both the Y and Xq17-q21 heterochromatin (Fig. 1a and Supplementary Fig. 7). Thus, heterochromatic status of these regions may need revision once we learn more about the *ETSTY7* sequence and functions. We also identified short eMSY sequences that showed high similarity to autosomes (Supplementary Note 1 and Supplementary Fig. 2). Autosomal transposition is a recognized mechanism for novel gene acquisition in MSY of other mammalian species[5,13,24] and apparently played a role in shaping the eMSY. Because of these diverse mechanisms, a recognizable portion of the horse Y chromosome has high sequence similarity with other chromosomes.

**Horse Y chromosome gene content and evolutionary origins.** A multipronged annotation approach resulted in identification of 52

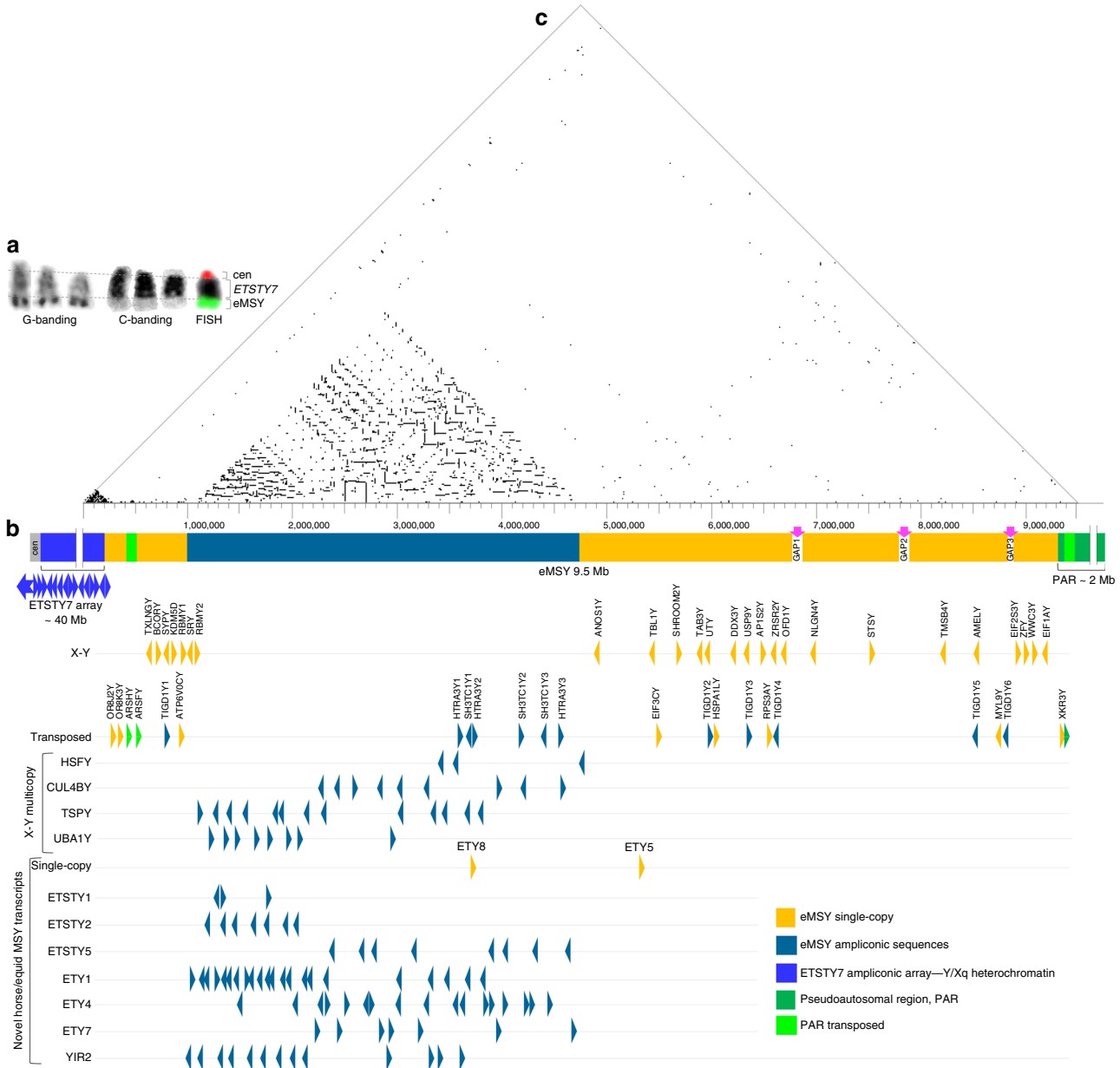

**Fig. 1** Horse Y chromosome organization. **a** Cytogenetic features of the horse Y chromosome: G-band positive material corresponds to MSY (green FISH signal); C-band positive material corresponds to *ETSTY7* ampliconic arrays; centromere is denoted by a red FISH signal (from published molecular profile of horse Y[68]); **b** Horse MSY sequence classes and gene map; **c** Triangular dot-plot showing the location of 100% identical sequences in MSY (with a 100 bp motif and a 20 bp step)

genes/transcripts of which 19 were not reported earlier[18]. Among these, 37 were single-copy and 15 ampliconic, representing cumulatively 174 gene/transcript copies (Table 1 and Supplementary Data 6). The gene density with 5.5 genes/transcripts per Mb and 18.3 gene/transcript copies per Mb was in the lower range of equine autosomes (5.4 genes/Mb to 28.2 genes/Mb) and closer to that of the X chromosome (8.3 genes/Mb)[17]. This is consistent with MSY annotations in primates[5,6,8] and pigs[14] and contrasts with the assumptions made prior to the genome sequencing era that the mammalian MSYs are gene poor and functionally inert.

The eMSY genes encompassed three main categories depending on their evolutionary origin: (i) X–Y genes (gametologs; 29), (ii) transposed genes (13), and (iii) Y-born horse-specific transcripts (10) (Fig. 1b and Table 1). In order to estimate the timeline for eMSY gene acquisition, we reconstructed individual

gene trees by aligning the eMSY mRNA sequences with equid and mammalian homologs (Supplementary Data 7).

The eMSY contains 29 X–Y genes, the highest number reported to date among eutherian Y chromosomes[1,8,10,12–14]. Notably, two of these, *TAB3Y* and *SYPY*, have ancient divergence times from their gametologs (123.5 MYA and 115.7 MYA, respectively) and are not Y-linked in any other species studied thus far, indicating that eMSY has retained a unique set of ancestral genes (Supplementary Table 3). The *WWC3Y* gene is also unique to horse MSY, but its intermediate divergence time (56.9 MYA) and position near the PAR suggest a more recent expansion of the MSY. In addition, because of a relatively short PAR in the horse, typical mammalian PAR genes, such as *ZBED1*, *TBL1*, *SHROOM2*, and *STS*[21,25], belong to eMSY in the horse.

The majority of horse X–Y genes were single-copy with broad expression in adult tissues (Supplementary Fig. 5 and

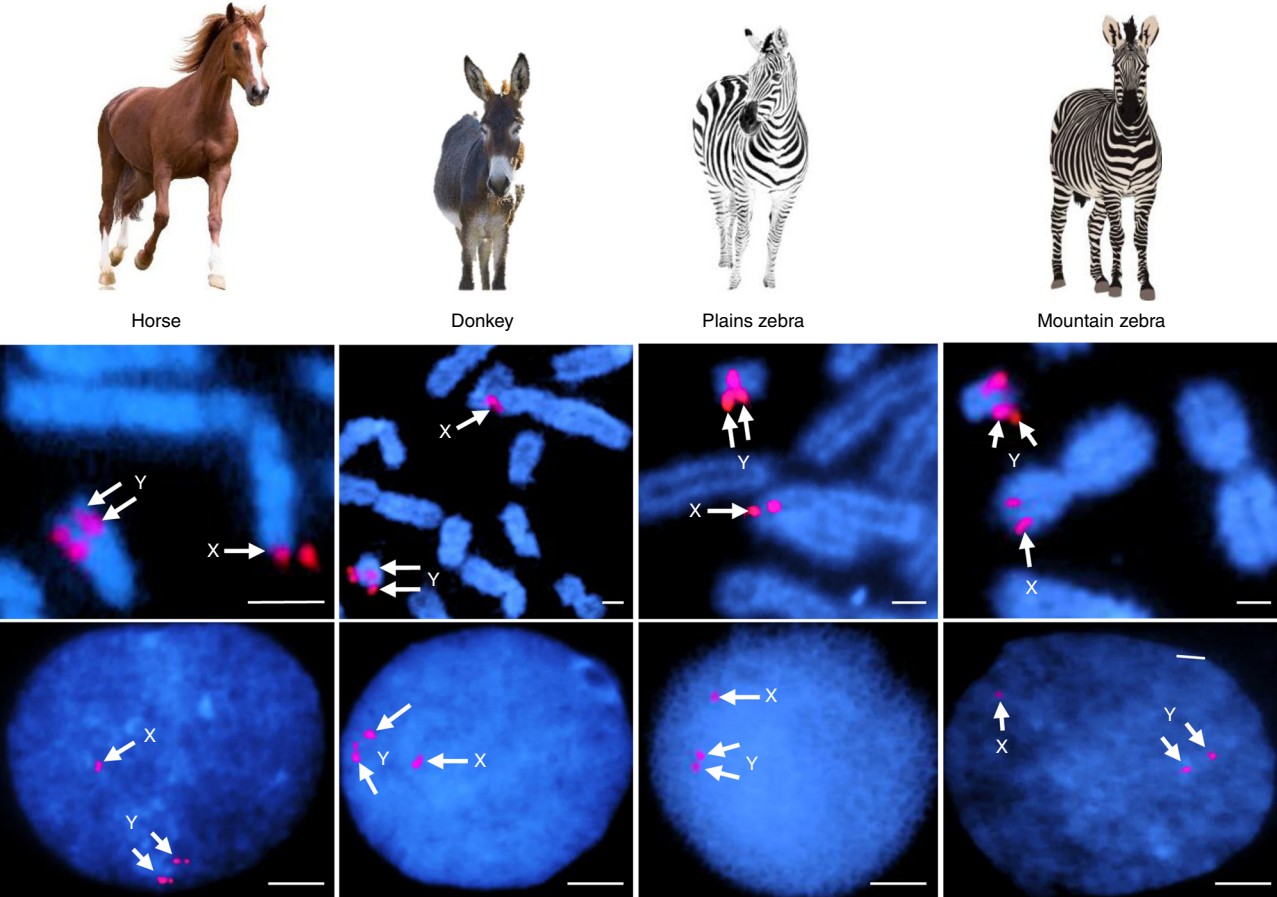

**Fig. 2** PAR transposition to MSY in horses and other equids. FISH with BAC 100H13 (see Supplementary Fig. 1) in metaphase (upper row) and interphase (lower row) chromosomes showing that PAR transposition to MSY has occurred in horses, asses and zebras; arrows indicate 2 distinct hybridization signals in the Y (corresponding to Y-PAR and MSY transposed regions) and one in the X chromosome (corresponding to X-PAR). Scale bar 1 μm. Equid images purchased from Bigstock (https://www.bigstockphoto.com/)

### Table 1 Summary data for eMSY genes and transcripts

| Sequence category | Genes | | | Expression profile | | | | |
|---|---|---|---|---|---|---|---|---|
| | Total | Single-copy | Multi-copy[a] | Broad | Testis dominant | Limited (tooth) | None | Tentative |
| Ancestral X–Y genes | 29 | 25 | 4 (33) | 17 | 8 (38) | 1 | 2 | 1 |
| Autosomal transposed | 10 | 7 | 3 (12) | 8 (15) | 0 | 0 | 2 | 0 |
| Y-born novel | 10 | 2 | 8 (92) | 2 (18) | 8 (76) | 0 | 0 | 0 |
| PAR transposed | 2 | 2 | 0 | 2 | 0 | 0 | 0 | 0 |
| Autosomal transposed to PAB | 1 | 1 | 0 | 0 | 1 | 0 | 0 | 0 |
| Total # genes | 52 | 37 | 15 | 29 | 17 | 1 | 4 | 1 |
| Total # copies/transcripts | 174 | 37 | 137 | 52 | 115 | 1 | 4 | 1 |

[a]Numbers in parentheses show the total number of gene copies in the category

Supplementary Data 6), which is a common feature of mammalian X–Y genes[12]. Four of the oldest X–Y genes became ampliconic: *HSFY*—3 copies, *UBA1Y*—8 copies, *CUL4BY*—9 copies, and *TSPY*—13 copies (Fig. 1b and Supplementary Data 6). However, compared to the massive amplification of *HSFY* in pigs[14,26] and cattle[12], or *TSPY* in cattle[12] and cats[13], their copy numbers in eMSY were moderate. The equine X–Y ampliconic genes were expressed in adult testis and, except for *UBA1Y*, in male embryonic gonads (Supplementary Fig. 5b), suggesting developmental functions prior to sexual maturity in addition to roles in spermatogenesis.

We noted that the horse *SRY* was a single-copy gene embedded between ampliconic sequences and surrounded by direct and inverted repeats (Fig. 1b). Such location facilitates *SRY* involvement in ectopic recombination within eMSY[27], resulting in *SRY* deletion and subsequently leading to disorders of sexual development such as the XY sex reversal syndrome[23]. Notably, XY male-to-female sex reversal counts for about 12–30% of all cytogenetic abnormalities in horses, whereas 70% of XY female horses have lost *SRY*[23]. In contrast, only 10–20% of human XY females (Swyer syndrome) have *SRY* mutations and the majority carry normal *SRY*, while the condition is rare or absent in other mammalian species studied, see ref.[23]. Thus, there are clear differences between species and we propose that this is due to MSY organization and, particularly, the location of *SRY*.

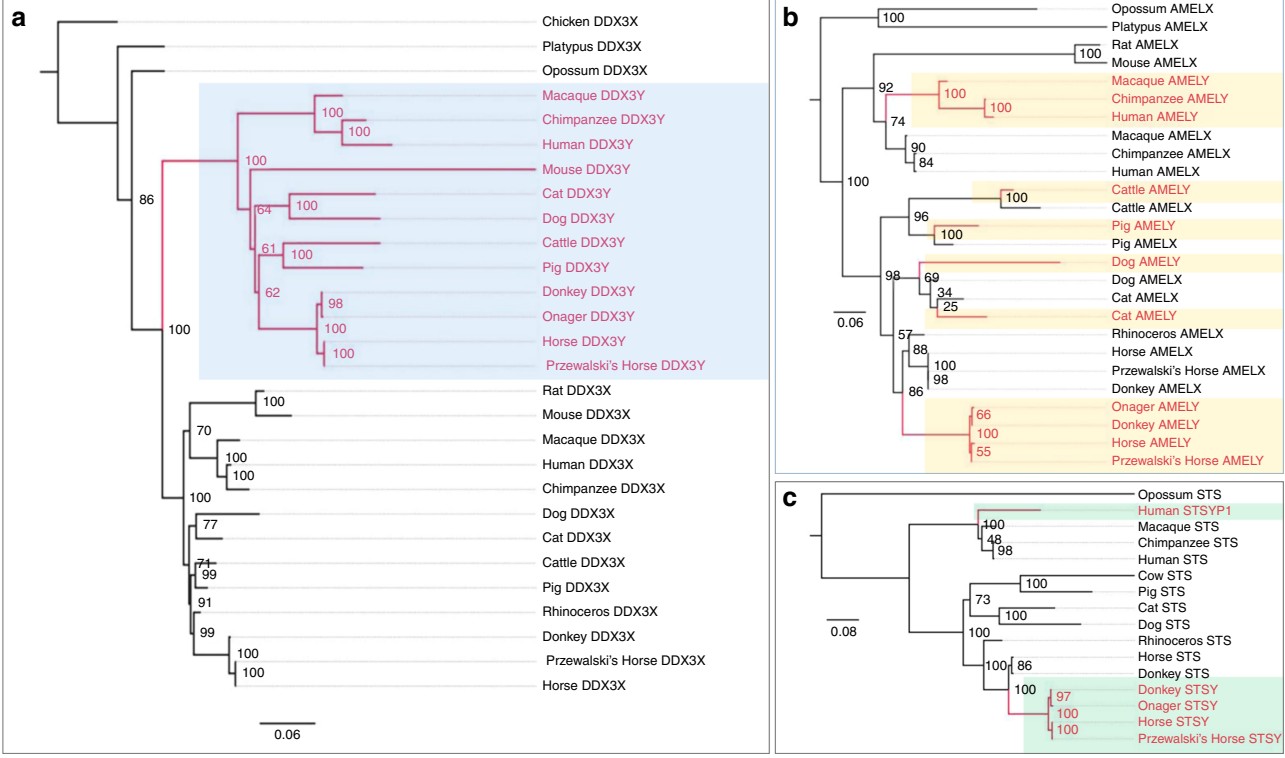

**Fig. 3** Evolutionary topologies of gametologs. Maximum likelihood phylogenies with bootstrap support values showing: **a** Monophyletic (e.g., *DDX3Y*) and (**b**-**c**) Polyphyletic (e.g., *AMELY*; *STSY*) patterns of evolution, with *AMELY* being broadly polyphyletic and *STSY* polyphyletic with two separate origins; Y orthologs in different species are highlighted with colored boxes

The horse Y genes were aligned with available X homologs to examine phylogenetic patterns and estimate the divergence times between gametologs. The topology of the X–Y genes fell into two categories (Fig. 3, Supplementary Table 3 and Supplementary Fig. 9). For 20 genes, all Y homologs formed a monophyletic clade consistent with a single origin in mammals (e.g., *DDX3Y*, Fig. 3a). Eight Y genes showed polyphyletic patterns in which some of the gametologs from the same mammalian lineage grouped together, suggesting parallel sequence conservation, potentially because of dosage compensation or gene conversion (e.g., *AMELY*; Fig. 3b). In others, the Y genes formed two distinct clades more consistent with multiple independent origins in placental mammals (e.g., *STSY*, Fig. 3c). Thirteen genes tested positive for gene conversion highlighting the importance of this process in the Y chromosome (Fig. 4; Supplementary Table 3 and Supplementary Data 3).

We assigned X–Y genes to evolutionary strata based on nucleotide divergence of synonymous sites ($K_s$) between equine gametologs and their location in the X chromosome, following Skaletsky et al. 2003[5] (Fig. 4, Supplementary Tables 3 and 5). For each gene, we also estimated the divergence time of homologs to approximate their acquisition in the Y chromosome. The oldest horse stratum corresponded to the therian *Stratum 1*[12,13] and contained four genes (*SRY*, *RBMY*, *HSFY*, *CUL4BY*) with a mean divergence date of 170.7 MYA and mean $K_s$ values of 1.18 (Fig. 4; Supplementary Table 5 and Supplementary Data 13). The second horse stratum was the same as *Stratum 2/3*[13] of other eutherians and contained 19 genes (Fig. 4b) with mean $K_s$ of 0.32 and mean divergence date of 105.8 MYA. However, this stratum had several outliers including *AMELY*, *ZFY*, *TMSBY*, *OFD1Y*, and *ZRSR2Y* that looked evolutionarily younger with depressed $K_s$ values compared to their position in the X chromosome. This is consistent with X–Y gene conversion after stratum formation as previously proposed for *AMELY* and *ZFY*[12] and in this study, also

for *TMSBY*, and *ZRSR2Y*. However, we did not detect gene conversion for *OFD1Y*, suggesting its more recent transposition. The youngest horse *Stratum 4* was unique to horse sex chromosomes and included *SHROOM2Y*, *TBL1Y*, *ANOS1Y*, *STSY*, and *NLGN4Y*, with mean $K_s$ of 0.13 and mean divergence date of 31.5 MYA. *Stratum 4* was adjacent to the PAR with their boundary demarcated by an autosomal transposed gene *XKR3Y*. This event was dated to 9.2 MYA, thus preceding the split of major *Equus* lineages[22]. Finally, the youngest X–Y region in the Y was a result of a transposition of *ARSHY* and *ARSFY* from the PAR (Figs. 2, 4) around 4.3 MYA coinciding with the emergence of the *Equus* genus[22], followed by gene conversion of *ARSFY*, which reduced the age for that gene to 2.3 MYA.

The complex phylogenetic patters of X–Y genes (Fig. 3, Supplementary Figs. 9–10) and their discordant $K_s$ values and divergence dates suggest that diverse evolutionary processes, such as chromosomal inversions and transpositions from the X and autosomes suppressed X–Y recombination in a stepwise fashion, creating evolutionary strata in the horse sex chromosomes that remained dynamic.

The horse MSY acquired 13 genes from various autosomal regions none of which have Y orthologs in other eutherians. The majority of autosomal transpositions (7 out of 10) occurred within the past 25 MY, with four very recent events occurring right around the divergence of horse from donkey (Fig. 4, Supplementary Table 3). The majority of autosomal transposed genes showed broad expression in adult tissues (Supplementary Fig. 5a)[18] and had a single-copy and reduced number of exons compared to their autosomal paralogs (Supplementary Data 6). The *SH3TC1Y* and *HTRA3Y* genes represented two outstanding exceptions. Both genes originate from the same region in horse chr3, both acquired 3 copies in eMSY and retained multiple exons in each. Sequence similarity between chr3 and eMSY was

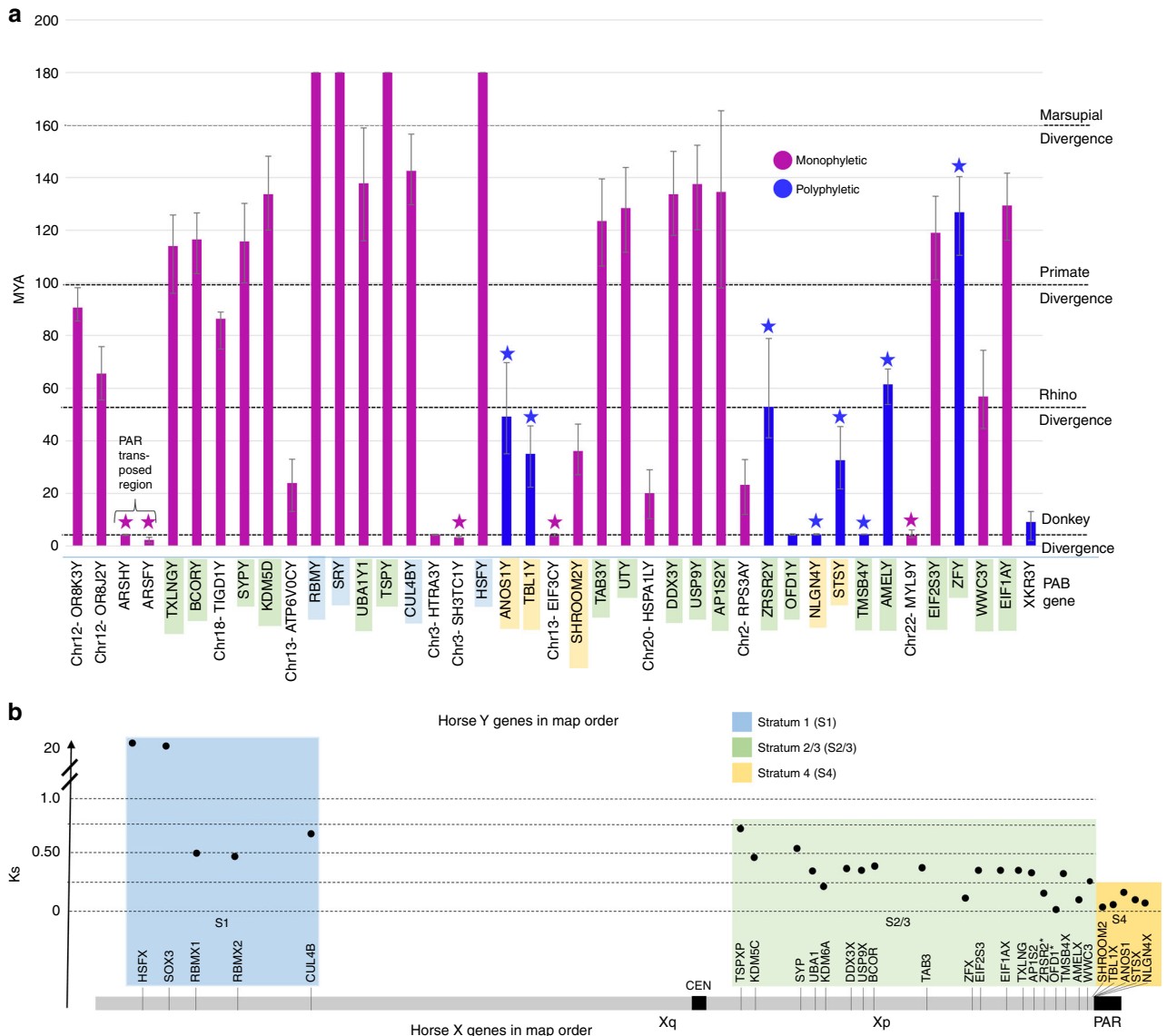

**Fig. 4** Horse MSY gene acquisition and evolutionary strata. **a** eMSY genes in map order are shown in x-axis; vertical bars in y-axis correspond to individual MSY genes, show their divergence time from X-gametologs/ autosomal paralogs, and are color coded according to phylogenetic patterns; stars above vertical bars indicate genes that tested positive for gene conversion; MSY gene symbols are color shaded according to their correspondence to evolutionary strata in the X chromosome as shown below; error bars show 95% Confidence Intervals estimated in *MCMCTREE*; **b** X-gametologs in map order are shown in x-axis and color coded according to evolutionary strata along the horse X chromosome; Ks values are plotted in y-axis

sufficient to be detectable by FISH (Fig. 5). This pattern, along with 1.7–2.2% divergence between *SH3TC1Y* and *HTRA3Y* paralogs suggests a relatively recent single transposition event around 3.3–4.1 MYA followed by gene conversion in *SH3TC1Y*. These genes are present in the multicopy region of the assembly and therefore there is some uncertainty regarding number of copies, structure, and orientation. Their broad expression pattern is noteworthy because other multi-copy Y genes are exclusively or predominantly expressed in testes. It is possible that because the recent origin, they have not yet acquired a testes related function. The recent gene acquisition events in the horse Y provide a unique opportunity to explore the consequences of increased gene dosage and structural rearrangements on the sex chromosomes.

Ten eMSY transcripts were considered novel Y-born and horse specific due to the absence of autosomal and X paralogs or orthologs in other eutherian species. As there were no sequences for comparison, it was not possible to estimate their time of origin. The majority of novel genes were ampliconic (Fig. 1), with 1–3 exons and expression in adult testis. Transcripts *ETSTY1*, *ETSTY2*, and *ETSTY5* were detected as early as 50-days post fertilization in embryonic gonads (Supplementary Fig. 5b). The function, if any, of these Y-born transcripts remain enigmatic, though some, like *ETY7*, represent novel equine noncoding RNAs (Supplementary Data 6) and may have regulatory roles in development or spermatogenesis, as recently shown for many tetrapod long noncoding RNAs[28]. Alignment of donkey testis transcriptome sequence with eMSY in this study, along with previous comparative analysis of eMSY genes in donkey[18] (Supplementary Note 5, Supplementary Fig. 6, and Supplementary Data 8), suggest that the majority of eMSY novel genes, except *ETSTY7*, have homologous MSY sequences in both equids. Some of these appear to be differentially expressed, like *ETSTY2*, which was abundant in horse but nearly absent in donkey testis transcriptome.

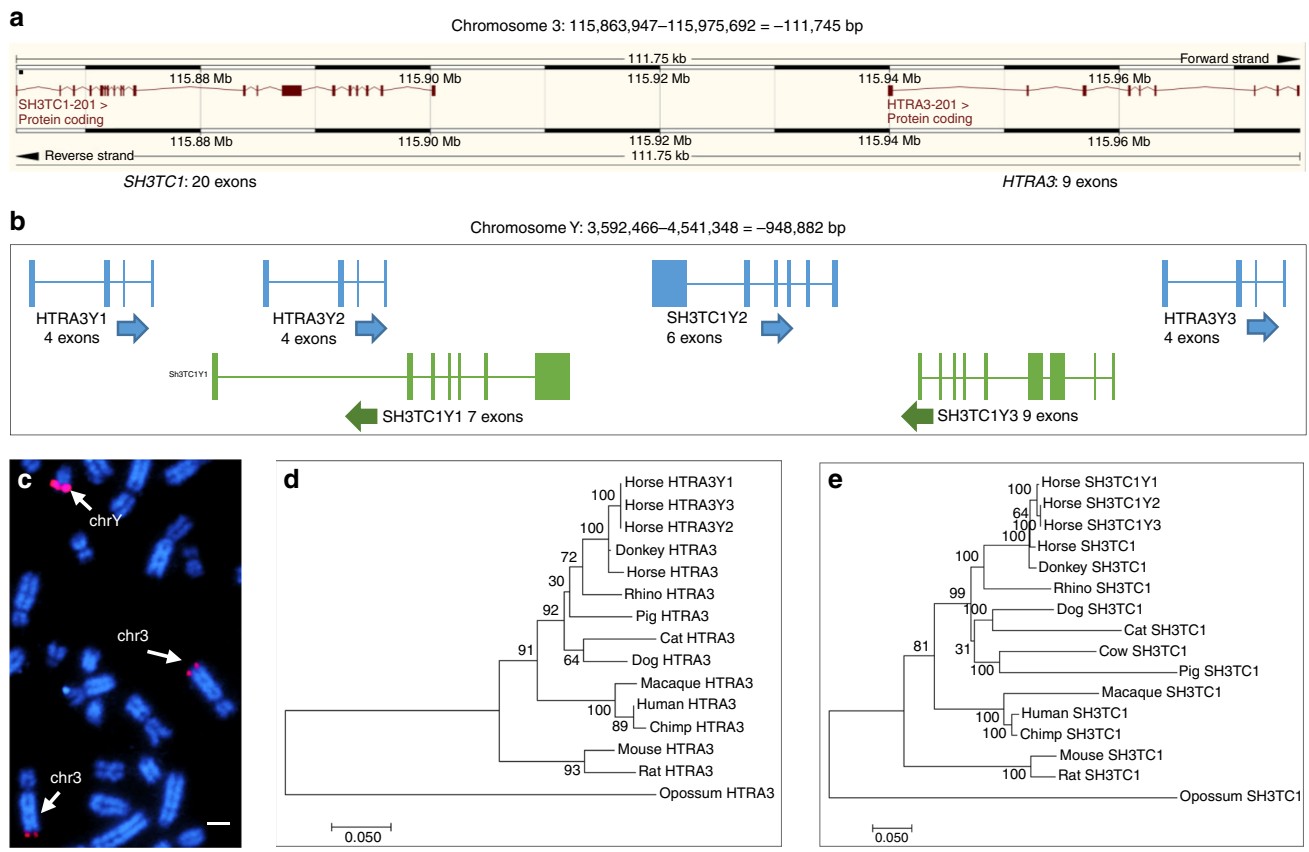

**Fig. 5** Acquisition of MSY genes by transposition from chr3. **a** Adjacent location of single *SH3TC1* and *HTRA3* in horse chr3 (http://www.ensembl.org/); **b** Close location of 2 copies of *SH3TC1Y* and 3 copies of *HTRA3Y* in eMSY sequence map; **c** FISH with eMSY BAC 139C20 containing *SH3TC1Y* and *HTRA3Y* in horse metaphase chromosomes; hybridization signals are present in eMSY and chr3qter (arrows); scale bar 1 μm; **d–e** Maximum likelihood phylogeny with bootstrap support values showing the clustering of three copies of *SH3TC1Y* and *HTRA3Y* in eMSY with their autosomal paralogs indicating recent divergence

**Amplification and horizontal transfer of a testis transcript *ETSTY7*.** The most notable among novel Y-born transcripts was *ETSTY7*. It showed testis-limited expression (Fig. 6)[18], had at least 15 copies with 3 exons each that collectively covered 50 kb of the most proximal 160 kb of eMSY (Fig. 1 and Supplementary Data 6). The size of *ETSTY7* repeat units ranged from 2.8 to 5.7 kb with a 9.8% average sequence divergence between the copies. FISH analysis with *ETSTY7* PCR product found additional massively amplified *ETSTY7* sequences throughout the Y heterochromatin, as well as in the facultative heterochromatin of Xq17-q21 (Fig. 6).

Ampliconic *ETSTY7* sequences were also detected in other equids but with different distribution patterns (Fig. 6 and Supplementary Fig. 7). In donkey, the sequences were in Xq only, suggesting that the low-level transcription of *ETSTY7* as previously observed by RT-PCR[18] and by RNAseq in this study, originate from X chromosome copies. In the zebras, *ETSTY7* sequences were in Y, Xpq, and the subtelomeres of several autosomes. No *ETSTY7* sequences were detected by FISH or PCR in the rhinoceros, an evolutionarily distant Perissodactyl species (Supplementary Note 6). We propose that the *ETSTY7* transcript family was acquired by the Equidae X chromosome after divergence from other Perissodactyls about 52–58 MYA[29], but before equids split 4.0–4.5 MYA[22]. Subsequently, the sequences were amplified and transposed to the Y in horses, and to the Y and autosomes in zebras (Fig. 6). Alternatively, it may be that *ETSTY7* copy number in donkey Y is too low for detection by FISH. Even though expansion of lineage-specific testis transcripts

is a characteristic feature of all studied mammalian Y chromosomes[5,10,13,14,26], *ETSTY7* distribution in equids is unique with no analogy in other eutherians.

Perhaps the most intriguing discovery was *ETSTY7* sequence similarity with the equine intestinal parasite *Parascaris equorum* (PEQ; genome assembly GenBank LM462759), better referred to *Parascaris* spp, because of difficulties delineating parasite species[30]. The region of homology included part of *ETSTY7* intron 2/3 (containing a LTR element) and exon 3, with 84–96% sequence similarity to multiple contigs in *Parascaris* genome assembly. To rule out the potential that these shared sequences were the result of the host horse sequences contaminating the parasite assembly, we extracted DNA from multiple individual adult worms, dissected adult tissues, L4 larvae and eggs (Supplementary Table 6) and designed primers from horse-specific, *Parascaris*-specific, and horse-parasite shared sequences (Supplementary Data 11). Only the horse-parasite shared-sequence primers amplified in both organisms (Fig. 6, Supplementary Note 6 and Supplementary Fig. 8).

We sequenced *ETSTY7* amplicons from multiple parasite samples and developmental stages and showed 90–95% similarity between horse and *Parascaris*. Multiple sequence alignment of horse *ETSTY7* and *ETSTY7*-like sequences from donkey, Przewalski's horse and *Parascaris* were used to reconstruct a phylogenetic tree (Fig. 6). The most divergent *ETSTY7*-like sequences were from the horse X chromosome and donkey unplaced scaffolds. The *ETSTY7* sequences of known equine Y origin formed a clade (97% bootstrap value), which also included

3 X copies, 7 Przewalski's horse sequences, and 6 *Parascaris* sequences. Of the latter, three were from GenBank and three were our sequences derived from whole worms, dissected gonads and body wall, and eggs and larvae (Supplementary Data 7). Clustering of Y copies and their divergence from the majority of X copies suggest they may be evolving separately. There were numerous *Parascaris* sequences placed across the tree, suggesting HT likely happened several times, while amplifications and transpositions in different equid lineages indicate *ETSTY7* as an active and mobile element (Fig. 6). Since *Parascaris* spp. is a

cosmopolitan intestinal parasite of equids[30] and *ETSTY7* sequences were abundant in donkey and zebra genomes, we cannot resolve when HT occurred.

Altogether, our data provide strong evidence of a putative HT between the horse and its parasite. Of immediate interest is to determine the origin of *ETSTY7* (e.g., transposon, noncoding RNA), and whether these sequences have any biological functions. Testis-specific expression alone provides limited clues because the permissive epigenetic regulation of testis allows transcription of many potentially nonfunctional sequences[31]. To

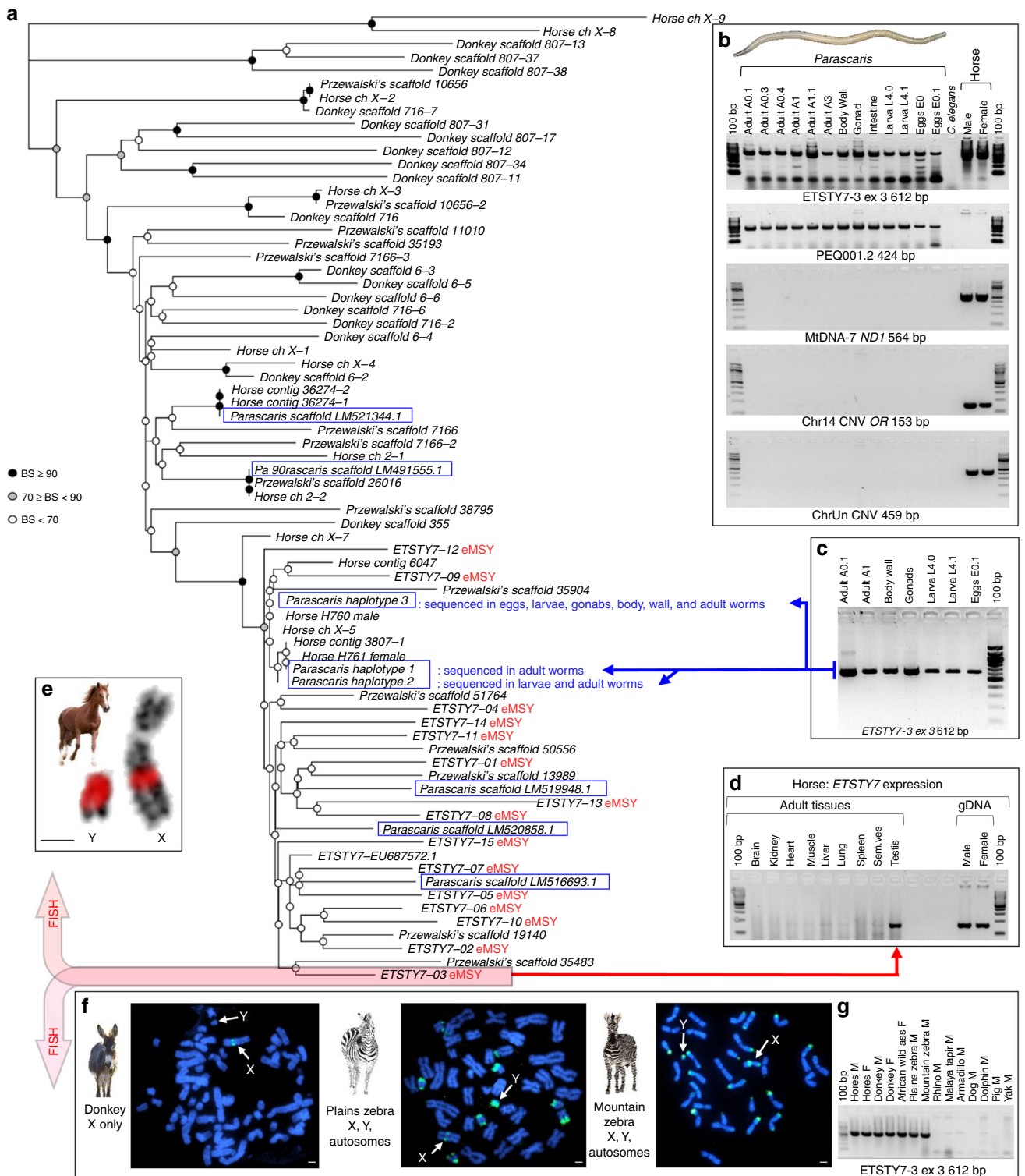

the best of our knowledge, this is the first HT described in equids, and among only a few verified HTs within vertebrates[32–34]. The exact mechanisms of eukaryotic HT are unknown, though host-parasite interactions are among the main known gateways[32]. It is possible that HT in eukaryotic genomes is more common than currently appreciated with more evidence emerging from the rapidly expanding genomic and transcriptomic data[34,35]. Unique male-specific ampliconic testes-expressed sequences have been identified in every Y chromosome studied this far, yet their origin has remained puzzling. Horizontal transfer of an ampliconic transcript we detected in equids suggests this may be one potential mechanism for the acquisition of novel genes.

**Candidate MSY genes for stallion fertility**. One of the main incentives for Y chromosome research in humans and domestic animals is that the Y accumulates sequences favorable for male biology and fertility due to hemizygosity and inheritance exclusively through the male germline (reviewed by ref.[15]). Studies of MSY mutations in humans[4,36] and mice[37–39] provide compelling evidence for this. However, except for corroborating that Y chromosomes are enriched with testis expressed genes, no such research has been done in other eutherians[12–16]. Here we generated testis RNAseq data for horse, donkey, and mule. Although some fertile F1 generation mules have been occasionally described[40], mules are generally sterile hybrids between a male donkey and a female horse. Thus, they carry the donkey Y chromosome. Sterility of mules is likely due to meiotic arrest during spermatogenesis or oogenesis[41], although other unknown genetic mechanisms may contribute[42].

We hypothesized that MSY genes with comparable expression in horse and donkey testis, but significant dysregulation in mules, may have roles in spermatogenesis or reflect more general genetic hybrid incompatibilities. Three-way RNAseq comparison revealed three genes that were significantly downregulated in mule testis: *HSFY* (66-fold), *HSPA1LY* (68-fold), and *XKR3Y* (395-fold) (Supplementary Fig. 6 and Supplementary Data 8). Of these, the multi-copy testis-specific *HSFY* is proposed as a candidate male-benefit gene in other eutherians. It belongs to critical azoospermia region (AZFb) in human MSY[43], and is massively expanded in the MSY of pig[26] and cattle[12]. Furthermore, *HSFY* has a chicken orthologue that is predominantly expressed in testis[12], suggesting that this is an ancient vertebrate male fertility gene, though *HSFY* biological functions remain currently enigmatic[26]. Even less is known about the functions of *HSPA1LY* and *XKR3Y*. *HSPA1LY* is a single-copy, broadly expressed, transposed gene whose autosomal paralog is a heat-shock protein and chaperone that facilitates protein folding. The *XKR3Y* gene shares sequence similarity with XK family of membrane transport proteins but has acquired testis-limited expression in horses (Supplementary Fig. 5). Functional dysregulation of these genes in mule testis suggests their role in

spermatogenesis and is supported by genome-wide gene ontology (GO) enrichment analysis of all other dysregulated genes in mule (Supplementary Note 5 and Supplementary Data 1). It is also possible that Y gene regulation and fertility phenotypes in equids depend on the numerical and functional balance between ampliconic gametologs—something similar to how the *Sly/Slx* loci regulate sex chromosome transmission and male fertility in mice[44]. However, current knowledge on such genes in horses/equids is too limited for speculations.

**Evolutionary dynamics of Y in ancient and modern horses and equids**. Besides stallion fertility, the Y is of interest for tracking male lineages and the history of breeds and populations using MSY sequence variants. Because eMSY assembly was not available until now, all previous population genetics studies have relied on relatively short MSY fragments. These studies unanimously indicate that the horse Y has experienced a drastic reduction in diversity following initial domestication some 5500 years ago[45–47], with a putative coalescence time of around 500 years[19]. In contrast, notable Y variation was present in ancient horses[48] and other extant members of *Equus*[49].

In order to gain more information about MSY variation, we mapped short-read whole genome data from 18 male horses and equids to our 9.5 Mbp eMSY assembly. The samples included eight modern horses of six breeds, four Yakutian horses, two ancient horses, two Przewalski's horses, a donkey, and an onager (Fig. 7 and Supplementary Data 9)[22,46,47,50]. After normalization, all samples mapped with similar (~1X) coverage to eMSY single-copy regions but showed a large drop in mappability to the ampliconic region (Fig. 7a and Supplementary Data 14). Such collapse in mappability was expected because the region is dense in sequences with almost 100% identity (Fig. 1c) and incompatible for mapping short reads. Further, there was an order of magnitude greater coverage in the first 0.25 Mbp of eMSY for all caballine (domestic and Przewalski's horse), but not asine (donkey and onager) samples (Fig. 7a). This is consistent with mapping *ETSTY7* arrays to this region in horses but not in donkeys (Fig. 6). Next, a small but prominent drop in coverage for all samples at around 0.3–0.4 Mbp coincided with PAR transposition, confirming that the transposition is present across equids (Fig. 2). Additional alignment of these sequences in the PAR explains the lower coverage. Finally, a region around 5.5 Mb showed increased coverage for the two ancient horses suggesting a duplication. We also noted a duplication in donkey at around 8.8 Mbp and a deletion in onager at around 5.8 Mbp, which remain tentative until confirmed in more individuals.

Following the initial mapping, we masked eMSY ampliconic sequences, removed ancient sample CGG10023 due to low coverage (Supplementary Data 9 and 14), and estimated the mismatch rate (Fig. 7b) and sequence variants (Fig. 7c). Donkey and onager were the most divergent from horse, consistent with

---

**Fig. 6** *ETSTY7* horizontal transfer between horse/equids and *Parascaris*. **a** Maximum likelihood tree for 73 *ETSTY7* and *ETSTY7*-like equid and *Parascaris* sequences (blue boxes) (sequence data Supplementary Data 7); copies verified to have originated from eMSY (red font) and sequences derived directly from horse and parasite samples noted; BS—bootstrap replicates; **b** HT validation by PCR using gDNA samples originating from 13 *Parascaris* individuals, tissues and developmental stages, *C. elegans* as negative invertebrate control, and male and female horses; Parascaris photo by R. Juras; primers represent the putative HT region (*ETSTY7*-3 exon3), *Parascaris*-specific sequences (PEQ001.2), and three multicopy horse-specific sequences - mtDNA (*ND1*), olfactory receptor genes in chr14 (Chr14 CNV *OR*), and a CNV in chrUn (ChrUn CNV); 100 bp ladder (NEB); **c** *ETSTY7*-3 exon3 612 bp PCR amplicons from *Parascaris* individuals, tissues and developmental stages that were used for Sanger sequencing; **d** Reverse transcriptase PCR (RT-PCR) with *ETSTY7* primers on a panel of 9 adult equine tissues showing testis-specific expression; **e**–**f** FISH with 612-bp *ETSTY7*-3 PCR product in horse sex chromosomes (red FISH signals in **e**) and metaphase spreads of donkey and two zebra species (green FISH signals in **f**); scale bars 1 μm; equid images purchased from Bigstock (https://www.bigstockphoto.com/), and **g** PCR with *ETSTY7*-3ex3 primers in equids, Perissodactyls and unrelated diverse mammals showing that these sequences are limited to equids; M male; F female (see Supplementary Data 11 for primer information and Supplementary Note 6 and Supplementary Fig. 8 for more HT validation)

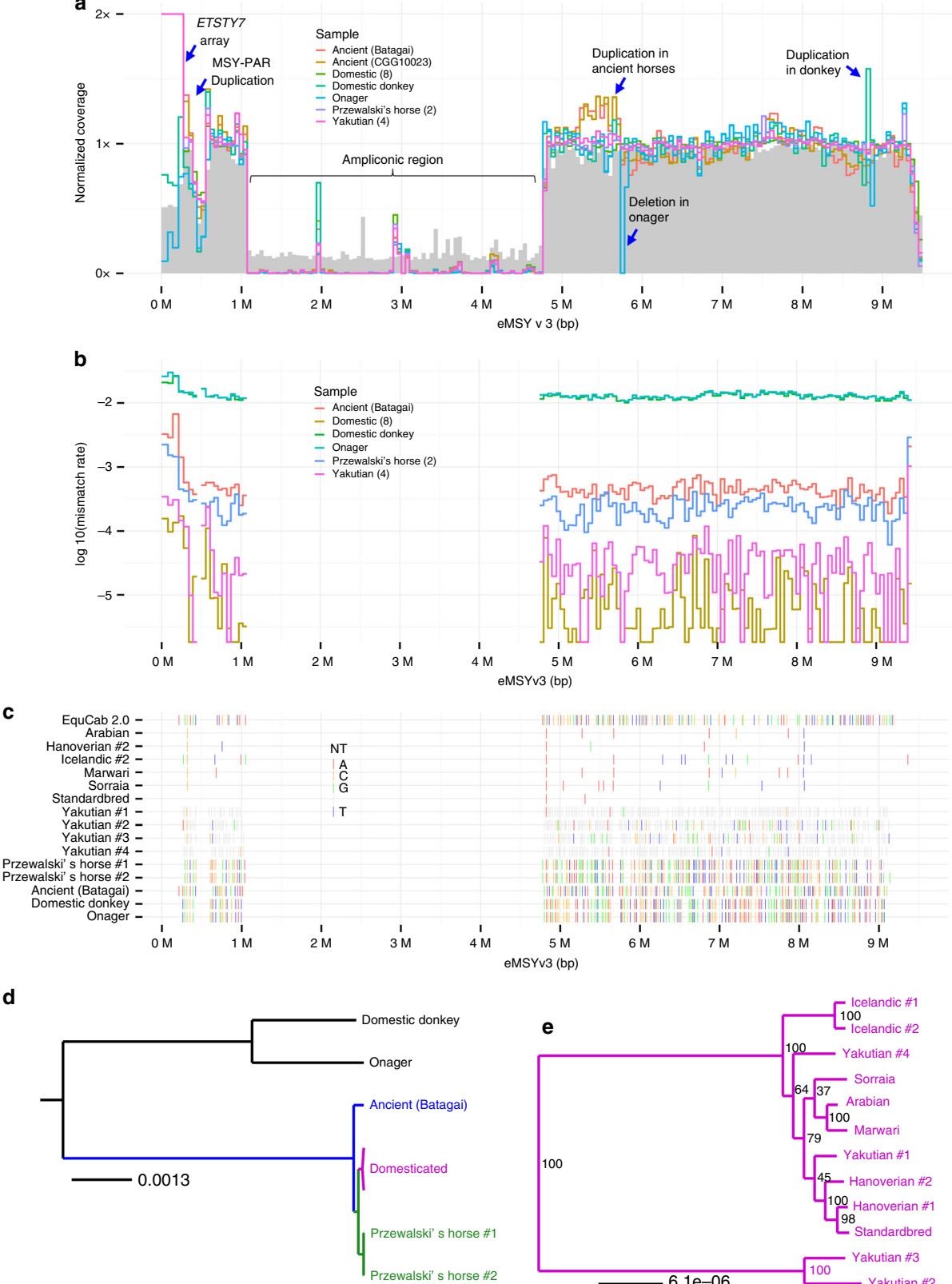

**Fig. 7** Comparison of equid MSY. **a** Normalized coverage for each group of samples in 1 kbp blocks. Mappability (between 0 and 1) is shown in gray. **b** Average mismatch rates (relative to reference sequence) in 1 kbp blocks. **c** Variable sites in eMSY. Only sites called in the eMSY sequence and covered in at least one individual in each group are shown. EquCab 2.0 shows the nucleotide observed on eMSY, while rows for individuals show the SNP observed at that site for an individual, or nothing if the reference nucleotide is observed. Gray indicates sites not called in that individual. **d** and **e** Maximum likelihood phylogenies generated from supermatrix of high-mappability genomic blocks for equids and horse breeds, respectively

the basal *Equus* divergence 4.0–4.5 MYA[22]. Mismatch levels were intermediate for the 5200 years old Batagai horse[47] and Przewalski's horses, and lowest for modern horse breeds. Mismatch and divergence analyses confirmed earlier findings that Batagai is equally distant from Yakutian and all modern breeds[46], and that MSY diversity was present in ancient horses but largely lost in modern horse breeds. However, these comparisons were limited to single-copy MSY and excluded the potentially more variable ampliconic region.

The alignment of eMSY with whole-genome short-read data from numerous horses and equids allowed reconstruction of MSY phylogeny for caballine and asine equids (Fig. 7d). The trees supported an early divergence of the ancient Batagai MSY, followed by a split between the Przewalski's and domestic horses. Among domestic horses, the most basal split separated two Yakutian haplotypes, followed by two Icelandic horses, which had the most divergent eMSY sequences among European breeds. Notably, Yakutian horses had substantially greater level of MSY diversity compared to the other domestic breeds[46]. The more divergent Y sequence in Icelandic horses suggested a unique breed history. The Standardbred horses also showed different ancestral haplotypes compared to those derived from the Arabian.

**The first representative Y assembly in Perissodactyla.** We present the first comprehensive MSY sequence assembly for the genus *Equus* and the first representative MSY of the eutherian order Perissodactyla. We compared the content of eMSY with recent most comprehensive comparative mammalian MSY studies[1,12,13] and added newly available data for pig Y[14,26]. This enabled a comparison across 12 species encompassing five eutherian orders (Perissodactyla, Primates, Carnivora, Cetartiodactyla, and Rodentia) and expanding comparison across metatherians by including data for opossum gametologs[12].

Among all sequenced eutherian MSYs, we recorded 88 unique genes and transcripts (Fig. 8 and Supplementary Data 10), of which 55% (49) were species-specific or lineage-specific. The highest number (23) of such genes was in eMSY, including several novel Y-born and autosomal transposed loci. The remaining 39 were X–Y genes and shared between species, allowing tracking lineage specific evolutionary events of acquisition, loss, and amplification. Notably, only six X–Y genes (*SRY, ZFY, TSPY, DDX3Y, UTY,* and *USP9Y*) are actively transcribed in all taxa, although a few were recently pseudogenized in one or two species. The trio, *DDX3Y-UTY-USP9Y*, as pointed out before[8,12,13,18], represents the only MSY conserved linkage group known to date. The horse MSY shared more X–Y genes with primates than with other eutherians, as a direct result of shorter PARs that evolved independently in these two lineages[21,25]. Among other shared X–Y genes were *BCORY* (horse-dog-pig), *AP1S2Y* (horse-pig), *ZRSR2Y* (horse-pig-cattle), and *CUL4BY* (horse-pig-carnivores) (Fig. 8).

One of the more surprising findings was that conservation of MSY in equids was much greater than expected based on the patterns observed in primates[6,7]. While the 4–4.5 MYA divergence time between horse and donkey[22] is comparable to the 6–7 MYA between human and chimpanzee[6], MSY evolutionary rate is different in the two groups. Over 30% of human and chimpanzee MSY sequence maps are not homologous and have substantially different gene content[6]. In contrast, horse and donkey retain nearly the same single-copy MSY and many of the multi-copy sequences, as revealed by short-read sequence alignment (Fig. 7a) and comparative gene expression analysis in this (Supplementary Fig. 6) and previous studies[18].

The 9.5 Mbp annotated sequence assembly of eMSY is currently one of the most complete MSY assemblies for any non-primate/non-rodent eutherian mammal, and provides additional insight into the dynamic nature of this chromosome and its heterogeneity between different mammalian lineages. The eMSY assembly also fills a primary gap in the current horse genome reference sequence, and establishes an important foundation for future research into stallion biology. Our analysis showed that while the eMSY shares organization and transcriptional patterns common to all eutherian Y chromosomes studied to date, it also exhibits novel features. Distinctive features of the eMSY included the largest retention of X–Y ancestral genes of any eutherian mammal studied thus far, and a unique collection of Y-borne and transposed genes. The unprecedented horizontal transfer between the horse Y and equine parasite, coupled with testis expression and amplification of this sequence in Y and X heterochromatin, provides a potentially novel model for the studies of host-parasite genome interactions and adaptation.

## Methods

**Ethics statement.** Procurement of blood and tissue samples followed the United States Government Principles for the Utilization and Care of Vertebrate Animals Used in Testing, Research and Training. The protocols were approved by Institutional Animal Care and Use Committee as AUP 2012–076 and CRRC09–47.

BAC tiling path map: (Supplementary Note 1, Supplementary Figs. 1–3 and Supplementary Data 4–5). We constructed a BAC contig map of eMSY by sequence tagged site (STS)-content analysis, chromosome walking, and fluorescence in situ hybridization (FISH) using methods described elsewhere[18,51,52]. Briefly, we designed primers for known and new eMSY markers with Primer3 software[53] and screened the CHORI-241 male horse BAC library (Thoroughbred "*Bravo*") (http://bacpacresources.org/) by PCR. If no clones were found in CHORI-241, we also screened the TAMU (L. C. Skow, unpublished) and INRA[54] BAC libraries, constructed from a male Arabian and a male Selle Français, respectively. The BAC DNA was isolated with Plasmid Midi Kit (Qiagen) and Sanger end-sequenced for STS development using standard T7 and SP6/M13 primers and BigDye chemistry. We used blood lymphocyte and/or fibroblast cultures and standard procedures[52] to obtain metaphase, interphase and DNA fiber preparations from a Thoroughbred stallion (*Bravo*), and metaphase and interphase preparations from a male donkey (*Equus asinus*; EAS), the quagga plains zebra (*Equus burchelli*; EBU) and Hartmann's mountain zebra (*Equus zebra hartmannae*; EZH). We labeled DNA from individual BAC clones with biotin-16-dUTP and/or digoxigenin-11-dUTP by nick translation (Roche) and hybridized the BACs individually or in combinations of 2 or 3 to metaphase/interphase chromosomes and DNA fibers. We analyzed a minimum of 10 metaphases and 30 interphase or DNA fiber-FISH images per experiment using a Zeiss Axioplan2 fluorescent microscope and Isis v 5.2 (MetaSystems GmbH) software. We arranged the BACs into a tiling path by STS content mapping (PCR with all STS primers on all BAC clones) and determined the order and orientation of contigs and the size of gaps by interphase and/or fiber-FISH. The BAC contig map formed the basis for eMSY sequencing.

Horse MSY sequencing and assembly: (Fig. 1, Table 1; Supplementary Note 2, Supplementary Fig. 1 and Supplementary Data 4–5). We sequenced 94 BACs from the BAC tiling path map using a multi-platform strategy which included 454 GS-FLX Titanium (Life Sciences) and MiSeq (Illumina) approaches. *Single-end (SE) 454 GS-FLX* was applied to 45 single-copy BACs from contigs Ia, Ic, II, III, IV with approximately 1.5X tiling path. Clones for each of the contigs were pooled respectively and sequenced to a depth of 30X. Contigs III and IV were short (~1 Mb), did not appear to share any sequences, and were pooled together. From the multi-copy region of contig IIb, we sequenced 11 BACs that represented all classes of sequence and content (based on STS content analysis): 3 BACs were sequenced individually, four were pooled into pairs, and four were pooled together. The pools combined clones that did not have any known shared sequence, thus increasing chances of assembling sequences unique to each BAC. *Large insert (6 kb) mate-pair paired-end (PE) 454 GS-FLX* was applied to build large scaffolds of 45 single-copy and 11 multi-copy BACs, as above. *Short read 2 × 250 bp PE Illumina MiSeq* was applied to 65 clones that were individually barcoded and sequenced in pools. *Long-read PacBio* was conducted in 2 SMRT cells—one containing a pool of 18 single and multi-copy BACs from contigs Ia and Ib, another with 40 clones from contigs Ic, II, III, and IV. Sequencing depth per BAC was 300X and 150X, respectively.

We used *Newbler* (454 Life Sciences) for de novo assembly of the 454 data based on respective BAC pools. Assembly parameters were optimized to yield the longest contigs and scaffolds. All scaffolds and contigs were aligned with eMSY STS markers (Supplementary Data 4) by BLAST (https://blast.ncbi.nlm.nih.gov/Blast.cgi) to further orient and assemble scaffolds/contigs based on the physical map. We used *Velvet*[55] for de novo assembly of the Illumina data and selected parameters yielding the longest contigs and scaffolds. For BACs sequenced on 454 and Illumina platforms, we conducted both *Newbler* and *Velvet* assemblies for comparison. We aligned and compared multi-copy BACs in *Mauve*[56]. Due to the complexity of multi-copy sequences, it was difficult to create scaffolds with high

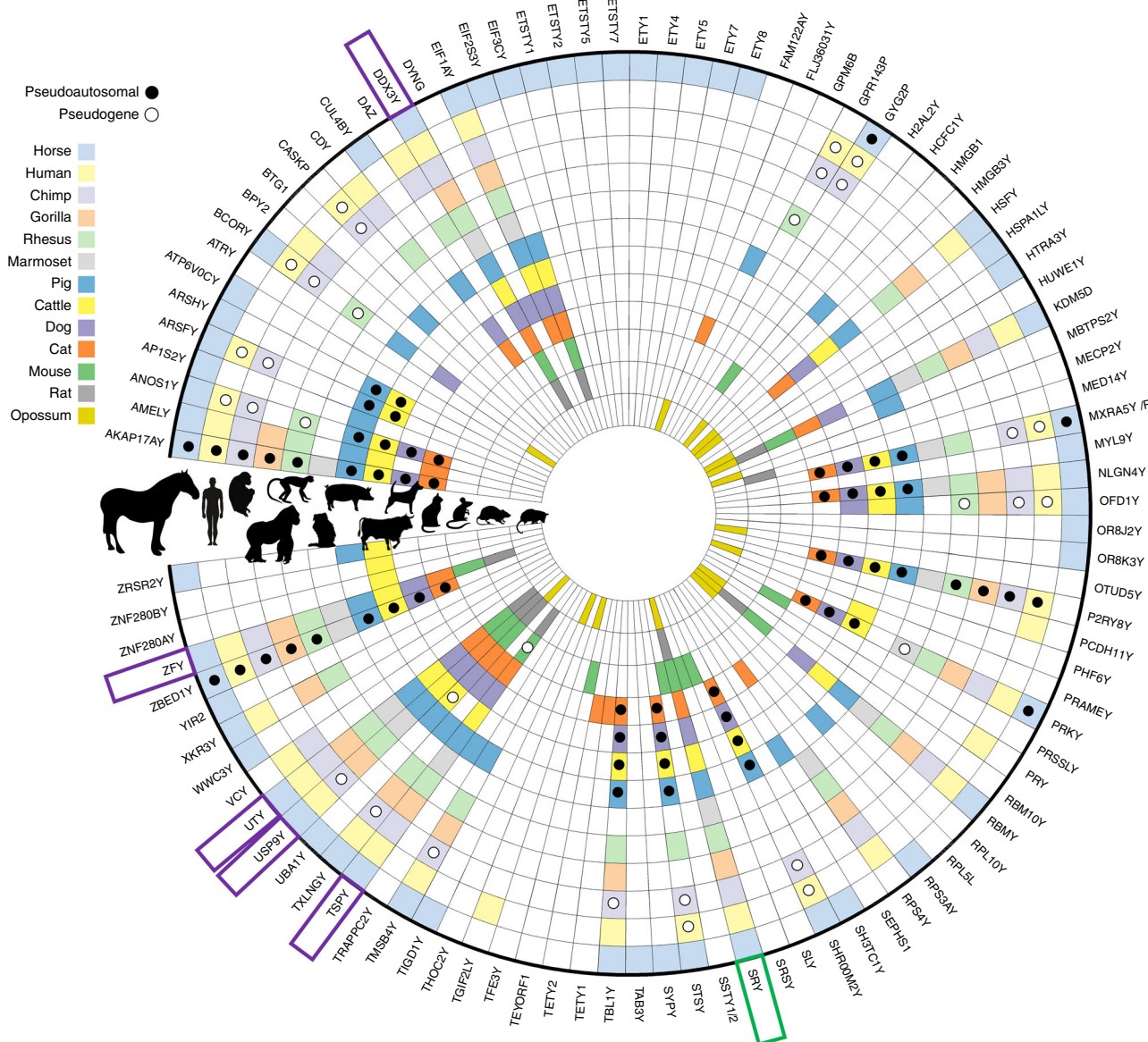

**Fig. 8** Comparison of MSY genes across Eutheria. Radial plot denoting the presence or absence of gene(s) on the Y chromosome of 13 mammalian species. Each concentric ring represents a species depicted in the silhouette on the left side, with the horse genes identified in this work as the outermost annulus. Colors corresponding to each species represent documented presence, and white indicates no current published evidence for the labeled gene. Pseudogene (ps) or pseudoautosomal (PAR) status of a gene is indicated by white or black circle, respectively. Five genes indicated by a purple box are present across Eutheria, and one indicated by a green box, across Metatheria (see Supplementary Data 10 for MSY comparative details). Images of mammalian silhouettes purchased from Bigstock (https://www.bigstockphoto.com/)

confidence, and we concatenated all representative BACs into one large scaffold. Therefore, the multi-copy portion of the eMSY assembly remained tentative. Attempts to incorporate the PacBio data with Illumina data using *Celera* Assembler (discontinued and replaced by *Canu*: http://canu.readthedocs.io/en/stable/) did not significantly improve the assembly and we dropped this strategy. Finally, we experimentally validated by PCR the assembly of selected gene containing regions (Supplementary Data 11) and considered amplification of the expected size products as proof of assembly correctness.

**Testis RNAseq.** We extracted high quality (RIN > 9.6) RNA from the testis of two normal adult stallions, two normal adult donkeys and two normal adult mules using PureLink RNA Mini Kit (Ambion). The RNA was converted into cDNA, prepared into 2 × 100 bp PE TruSeq libraries (Illumina), and sequenced on HiSeq (Illumina) platform. We obtained, on average, 80 million PE reads per sample and used *Trinity*[57] to assemble transcriptomes of the horse, donkey and mule testis by combining data from two individuals per species.

Sequence annotation: (Supplementary Data 2 and Supplementary Note 3). The eMSY sequence was analyzed for GC content and the content of interspersed

repeats in RepeatMasker (http://www.repeatmasker.org/) (Supplementary Table 2). Sequences with 100% intra-chromosomal identity were revealed with a custom Perl code (similar to that applied to human MSY[5]) and BLAST (http://blast.wustl.edu/) (Fig. 1c; Supplementary Note 3 and Supplementary Fig. 4). Annotation of eMSY genes and transcripts was a combination of rigorous bioinformatics analysis against: (i) known horse MSY genes[18] and STSs (Supplementary Data 4), (ii) available animal cDNA and EST databases, and horse, donkey and mule testis transcriptomes. First, we downloaded and analyzed by BLAST against eMSY assembly all available sequences (genomic fragments, mRNAs, ESTs) of mammalian MSY genes from Ensembl (http://www.ensembl.org/) and NCBI (https://www.ncbi.nlm.nih.gov/). Annotation of genes followed rigorous criteria: there had to be at least 80% sequence similarity, presence of at least 75% of the exons, and conservation of exon order and exon-intron boundaries. We excluded highly repetitive and fragmentary alignments. Next, we combined the annotated eMSY to the horse reference genome EquCab2 (https://www.ncbi.nlm.nih.gov/genome/?term=horse) and mapped horse testis transcriptome to it (Supplementary Data 8). This confirmed and refined the gene models obtained by BLAST, and identified novel genes and transcripts. Finally, we used in silico annotation with *Maker*[58] combining all horse EST, mammalian MSY cDNAs and horse testis

transcriptome (RNAseq assemblies) data. We validated all annotations by BLAST or BLASTP using the closest homolog (https://blast.ncbi.nlm.nih.gov/Blast.cgi/).

Gene expression analysis: (Supplementary Note 4). Expression profiles of all newly discovered genes, i.e., those not reported in the first horse MSY gene catalog[18], and selected known ampliconic transcripts, were determined by reverse transcriptase PCR (RT-PCR) in 9 adult male tissues and 9 tissues of 50-day-old male embryos. The adult tissues included brain, kidney, heart, skeletal muscle, liver, lung, spleen, seminal vesicle, and testis. Embryonic tissues included brain, kidney, heart, liver, lung, GI tract, chorio-allantois, gonad, and gubernaculum. All tissues were preserved in RNAlater (Ambion). We isolated RNA with RNeasy Plus and RNeasy Lipid kits (Qiagen) and converted it into cDNA with Verso cDNA Synthesis kit (ThermoFisher) following manufacturers' protocols. Tissue-specific cDNA together with male and female horse genomic DNA controls, served as templates for PCR reactions with primers specific to eMSY genes/transcripts (Supplementary Data 11 and Supplementary Fig. 5).

Raw reads from the six RNAseq libraries (2 horses, 2 donkeys, 2 mules) were aligned to the hybrid assembly of eMSY and the female horse reference genome EquCab2 (https://www.ncbi.nlm.nih.gov/genome/?term=horse). We used STAR two-stage aligner[59] and non-stringent mapping parameters (SI) to allow divergent read mapping across species. Read counting was performed with the Python framework HTSeq (10.1093/bioinformatics/btu638). Per-individual read coverage per gene is presented in Supplementary Data 8, inter-species comparison in Supplementary Data 12, Supplementary Fig. 6 and Supplementary Table 4, and genome-wide comparison in Supplementary Data 1.

**Validation of horizontal transfer (HT)**. Analysis by BLAST (https://blast.ncbi.nlm.nih.gov/Blast.cgi/) aligned *ETSTY7* transcript sequence with *Parascaris equorum* (*Parascaris* spp.) genome (GenBank LM462759). To validate putative horizontal transfer and check for possible contamination, we conducted a rigorous series of experiments. Using established protocols[30], *Parascaris* spp. eggs, larval stage 4 (L4) and adult specimens were collected from affected horses. We dissected adult gonads, body wall and intestine, and isolated genomic DNA from all developmental stages and organs using standard phenol-chloroform protocol. We designed primers from 3 types of sequences: (i) Parascaris-specific sequences; (ii) horse-specific MSY, autosomal, and mitochondrial DNA sequences, and (iii) horse-parasite shared sequences (see Fig. 6, Supplementary Note 6, Supplementary Data 11 and Supplementary Fig. 8). We sequenced *ETSTY7*-3 ex 3 amplification products from adult worms, their gonads, body wall, eggs, and L4 larvae by BigDye chemistry, and estimated the rate of parasite and eMSY sequence homology by BLAST.

**Phylogenetic trees and estimation of gene acquisition**. Gene sequences for X and autosomal paralogs and Y homologs were obtained from the mRNA RefSeq database (https://www.ncbi.nlm.nih.gov/) for representative eutherian mammals, opossum and platypus. Outgroup sequences from the chicken were also obtained if a clear homolog was present. Przewalski's horse, donkey, and onager Y sequences were obtained by mapping whole genome shotgun sequences from males to the eMSYv3 and extracting the sequences based on our annotation. Sequences were aligned using *MUSCLE*[60] and phylogenies reconstructed in *RAxML*[61] with the Maximum Likelihood algorithm using the General Time Reversible + Gamma + Invariant sites and significance evaluated with 1000 bootstrap replicates. Nodes in the best scoring ML tree for each gene were constrained using the 95% CI interval estimates (Supplementary Data 13 and Supplementary Table 5) from (http://www.timetree.org)[62] and the previous estimate of 4.0–4.5 for MRCA of *Equus*[22]. The divergence times and 95% CIs of the horse Y gene from its nearest autosomal or X homolog were estimated using independent rates model and soft constraints in *MCMCTREE*.[63](Supplementary Data 7 and Supplementary Figs. 9–10). For 5 genes the program did not produce 95% CI intervals and so they were approximated from the mean of 95% CI of other genes in their age range. The *yn00* program in *PAML*[63] was used to estimate the $K_s$ between the horse Y and X gene-pair using the method by Li et al. 1993[64]. Test for gene conversion was performed in the program *GENCONV*[65] (Supplementary Data 3).

**Horse MSY comparison in equids and ancient and modern horses**. Published short-read (Illumina) sequence data for 18 equids[22,46,47,50] (Supplementary Data 9) were mapped to the eMSY sequence using previously described procedures[46] for mapping reads, and for mapping to Y chromosome fragments, specifically. We analyzed the data in the context of MSY sequence coverage, mismatch rate, haplotypes, and phylogeny. Note that the ancient horse CGG10023 had a low coverage (max 7.4X; min < 4X) and poor quality sequence (average mismatch rate 3-fold higher than Batagai), and we excluded this sample from plots that are sensitive to the coverage, i.e., everything but the average coverage plot; Fig. 7a.

Average coverage for each sample was calculated in blocks of 50 kbp, considering only sites called in the eMSY reference sequence. Consequently, each physical block could cover more than 50 kbp of the eMSY reference sequence. The resulting averages were normalized using the mean coverage of high-mappability blocks.

Mappability was calculated by running the GEM Mappability v1.315 tool on the eMSY sequence using 100 bp k-mers (http://algorithms.cnag.cat/wiki/The_GEM_library). Scores were converted to WIG format, and the average

mappability of each 50 kbp genomic block was calculated, considering only sites covered in the eMSY sequence (see Supplementary Data 14).

The full eMSY sequence of each sample was genotyped as described in Schubert et al. 2014[47]. Thereafter, mismatch rates were calculated for each sample as the fraction of called sites containing non-reference nucleotides, grouped into 50 kbp blocks (see Coverage), considering only high-mappability blocks. The results were averaged across groups, to produce the final mismatch rates.

Genomic blocks were demarcated as described for Coverage statistics, and 1 kbp blocks with an average mappability greater than or equal to 0.9 were selected. The resulting genomic blocks were genotyped and phylogenetic trees were calculated from a super-matrix of these sequences as described by Schubert et al. 2014[47].

**Comparative analysis with eutherian MSY**. We compiled data for all MSY coding genes, pseudogenes and transcripts known in horses (this study) and eutherian species with sequenced or partially sequenced MSYs: man[5], chimpanzee[6], gorilla[11], rhesus macaque[7], marmoset[8], pig[14,26,66], cattle[8,12,67], dog and cat[13], mouse[8,10] and rat (https://www.ncbi.nlm.nih.gov/; http://www.ensembl.org/). To expand phylogenetic scope, we included data for opossum X–Y genes[12]. The comparative data were visualized with Circos (http://circos.ca/).

**Data availability**. Sequences and metadata generated in this work are publicly available. *BioProject Accession PRJNA420505* includes information about project, samples, and SRA accessions including *SRR636826-SRR636831* for RNAseq reads of 6 samples from 2 horses, 2 donkeys, 2 mules) and *SRR6361136-SRR6361138* (for 54 single-end reads and mate-pair reads for tiling path BACs, and Illumina paired-end reads of individually barcoded multi-copy BACs. The eMSYv3.1 assembly has been deposited in GenBank under accession number MH341179. In the v3.1 accession, two vector sequences (18,381-bp in position 2,505,950 and 1396-bp in position 8,066,811) have been trimmed from assembly with respect to v3.0. The GTF file for v3.0 available as Supplementary Data 2 with information on how to modify v3.1 assembly to match the v3.0 GTF annotations used in our analysis. GenBank accessions to BAC end sequences, MSY STS sequences and Sanger sequences of *Parascaris* samples can be found in Supplementary Data 4 and 7.

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

## Acknowledgements

This work was supported by USDA (grant 2012-67015-19632, BPC, TR, JEJ), LINK Endowment Equine Foundation (BPC, TR), Danish National Research Foundation (grant DNRF94, LO, MS), Initiative d'Excellence Chaires d'attractivité, Université de Toulouse (OURASI, LO), the European Research Council (grant agreement ERC-CoG-2015-681605, LO, MS). We thank Dr. Loren Skow and Dr. François Piumi for providing BAC clones from TAMU and INRA BAC libraries, respectively; Dr. Doug Antzack and Dr. Donald Miller for providing blood samples and fibroblast cell lines from the Thoroughbred stallion *Bravo*; Elaine Owens for assisting with cell cultures; Dr. Vaishali Katju for providing DNA from *C. elegans*; Dr. Rytis Juras for the photos of a *Parascaris* specimen. Special thanks to Dr. James MacLeod from the University of Kentucky for bonding Y chromosome researchers with equine parasitologists.

## Author contributions

B.P.C. initiated and designed the study. B.P.C. and T.R. supervised the work and coordinated collaborations between groups. J.E.J., T.R., C.D.J. and R.P.M. produced sequence data. J.E.J., T.R., B.W.D. assembled and annotated the sequence. T.R., J.E.J., N.P., P.J.D., S.G., R.P.M., C.D.J. and A.M.A.Z. conducted the laboratory work. T.R., D.D.V., C.C.L., and T.A.E.S. collected and phenotyped the samples. J.E.J., B.W.D., W.B. and G.L. conducted bioinformatics analyses and interpretations in close consultation with T.R., B.P.C., W.J.M. and D.W.B. M.K.N. collected, phenotyped, and processed *Parascaris* samples. T.R., J.E.J. and M.K.N. validated horizontal transfer. L.O. and M.S. conducted

comparative sequence analyses in equids. T.R., J.E.J. and B.W.D. produced the figures and wrote the text with input from all authors.

## Additional information

**Competing interests:** The authors declare no competing interests.

