## [Peer Review File · Nature Communications]

Reviewers' comments:

Reviewer #1 (Remarks to the Author):

Sequencing of mammalian Y chromosomes is a complex task, and one that is essential for our understanding of sex chromosome evolution. Janečka et al have generated the first horse Y chromosome sequence assembly, based on sequencing of ordered BAC clones. Their sequencing strategy was sensible and has given good coverage of both single copy and ampliconic regions of the eMSY, albeit with some caveats as to final clone orientations and positions. About 9.5Mb sequence has been produced from a ~45Mb chromosome, and since the chromosome is ~2/3 heterochromatin this represents a good proportion of the euchromatin.

Gene content of the Y was assessed by RNASeq, with gene structure additionally verified using genomic PCR on males and females, plus RT-PCR on selected tissues. This was valuable to confirm the gene annotation the authors performed, and supports the existing data on horse Y gene expression.

The authors have also carried out phylogenetic and evolutionary analyses of the sequences on the chromosome, most notably finding evidence for horizontal gene transfer from an intestinal parasite. Sufficient information is given to replicate the phylogenetic work, and the analyses are appropriate for this study.

This is a well written and interesting manuscript that fills an important gap in mammalian genomics, and I am happy to recommend it for publication. I have some specific comments below:

- 1) The comparative expression between donkey, mule and horse is interesting. The differential regulation of genes based on the genomic background is something that has been seen in mice - see for example the Six/Sly genomic conflict (PMID: 23028340). These ampliconic genes on the X and Y are competing to distort their transmission ratio, and one way this is mediated includes a broad regulation of sex chromosome expression. A phenotype only emerges when X and Y chromosomes from different backgrounds (with different copy numbers of the relevant genes) encounter each other, and no longer balance each other out. I wonder if a similar situation is involved on the horse sex chromosomes, and if the authors have already considered this?
- 2) The selected Y genes such as AMELY with low Ks described in stratum S2/3 should be tested for gene conversion, which may help identify the mechanisms at work.
- 3) Line 227 - '[the genes] have yet to acquire a testis related function'. I'd rephrase this as 'have not acquired a testis related function' - transposed genes are not guaranteed to gain novel functionality.
- 4) The ETSTY7 story: were any RT-PCRs carried out on donkey or mule to see if the X copies are expressed in testis as well as Y copies? The RNASeq data in figure S7 looks to show ETSTY7 sequences expressed in horse testis strongly, and a much lower level in donkey. Presumably the donkey transcripts must be coming from X copies?
- 5) Can the horse X and Y ETSTY7 copies be distinguished at the sequence level? That is, are they evolving in separate directions, which may indicate novel functionality?
- 6) The dot plots in Figure S4 suggest two diverging patterns to the repeat when comparing the 50bp and 100bp resolution; there are distinct blocks visible at 100bp which are otherwise similar at 50bp resolution. Do the underlying sequences give an indication of what has happened? For example, could this reflect intrachromosomal inversions mixing diverging sequences?
- 7) Within the supplementary tables S6, pig gene accessions refer to the Vega website; since pig genome 11.1 went live, the Vega gene accessions have been archived. It would be helpful to update the pig accessions to their live Ensembl IDs.
- 8) The autosomal transposed genes have a broad expression pattern. Is this also true of their autosomal paralogues?

9) Figure 4 is quite hard to interpret. It is difficult to track the stratum versus age colouring in panel A especially. Please try redrawing this figure; perhaps changing the bars to points, with a symbol representing the type of gene would be clearer.

Reviewer #2 (Remarks to the Author):

In the paper "Horse Y chromosome assembly displays unique evolutionary features, putative stallion fertility genes and horizontal transfer", Dr Raudsepp and co-workers present a first almost complete assembly of the male specific regions of the horse Y-chromosome. I really appreciate the work the authors performed to provide this comprehensive analysis. They not only show functional annotation, in addition they did multiple analyses to address very interesting aspects. This ranges from gene content on the horse Y, over prediction of horse MSY genes responsible for stallion fertility to evolutionary dynamics and lineage specific. This manuscript provides a comprehensive view on this particular part of the horse genome. It covers a very interesting topic which will likely gain great interest in the field.

The horse MSY assembly is highly awaited in the horse community and due to the widespread analysis performed, the manuscript will attract interest from geneticist and breeders also from other animal genome communities.

The manuscript is clearly written (some minor comments are pointed out below). The Authors provide detailed methodological information so that the experiments could be understood. Most claims are sound to me and appropriately discussed with regard to previous literature.

Overall the paper is a high quality study and I recommend the manuscript for publication except one aspect needs an additional test to improve reliability at this state in my opinion.

#Coming to my major point: Horizontal Gene Transfer from *Parascaris* to the eMSY

The authors first describe a horse Y-chromosomal ETSYT7 multi copy element. As the ETSYT7 element is present also in donkeys and zebras but not in the Rhino it must have its origin in the ancestor of the Equine lineage prior to the split of horses from donkeys/zebras about 4-5 million years ago. In horses ETSYT7 is detected on the eMSY and on the X and whereas it is found also on other chromosomes in other equids indicating mobility of the motif in the genome.

After observing homologous ETSYT7 sequences in an assembly of the horse intestinal parasite '*Parascaris equorum*' the authors hypothesize that the ETSYT7 sequence came into horses from the *Parascaris* genome via horizontal gene transfer.

This is a somehow unprecedented observation and the authors performed experiments to prove their hypothesis (Figure6). But I am not totally convinced that their experimental tests rule out a possible contamination of the worm DNA with horse that would refute their assumptions.

Regarding the *Parascaris* assembly: when I Blast the horse mtDNA sequence (NC_001640) to the *Parascaris* assembly (GCA_000951375.1), I find a perfect match of the horse mtDNA with '*Parascaris* >LM463828.1 *Parascaris equorum* genome assembly P_*equorum*, scaffold PEQ_contig0000001'. This means to me, that the *Parascaris* assembly is contaminated with horse sequences. The *Parascaris* assembly could therefore also be contaminated with horse ETSY7 elements and to my opinion it should be proven carefully, if the ETSY7 element is really embedded in the *Parascaris* genome.

The authors tested the occurrence of the ETSYT7 repeat in the *Parascaris* genome by PCR amplification of ETSYT7 using genomic DNA isolated from different worm tissues and worm developmental stages as template. The worms were collected from affected horses. In Figure 6A.1

the *Parascaris* control amplifies similar in all samples, whereas the ETSTY7 signal does not.

I think that the experiments shown in the manuscript cannot definitely confirm, that the PCR product shown for ETSTY7 (Figure 6 A, 3) derives from the worm genome and not from horse contamination.

The authors did only one experiment to test that ETSTY7 derives from the worm and not from horse DNA contamination. As a prove they use the absence of the horse Y-chromosome specific TSPY signal in the worm tissues (Figure 6A,2).

For my opinion TSPY is not an ideal control for horse DNA contamination in this context because
a) TSPY is male specific. In case of the worms were extracted from an affected female, TSPY would not amplify even in case of horse DNA contamination.

b) ETSTY7 is occurring in multiple copies on the X and the Y of the horse. Therefore PCR amplification is probably much more efficient for ETSTY7 than for the 13 copy TSPY gene (ETSYT7/exon3 primers give a much more prominent band than TSPY Fig6A).

My suggestion is to test the absence of horse DNA with a more appropriate (or several) horse DNA amplification controls. One could use horse multicopy autosomal loci or mtDNA instead of TSPY, to convincingly show that the ETSYT7 signal does not derive from horse contamination in the worm sample.

Also it would be good to see these tests (because of sensitivity, amplification efficiency, copy number estimate) in a qPCR experiment.

#Minor comments:

Unfortunately no pages, I therefore refer to the sections:

Section-Unique features of the horse Y chromosome

L103-111: Over half of eMSY is repeats.....

in L108 they write over 60% of the eMSY consists of single copy sequences. This is confusing.

Section- autosomal transposed genes

L215: ...with three very recent events occurring after the divergence of horse and donkey.

Which genes do you refer to here? I assume it is EIF3CY, MYL9Y and SH3TC1Y? But I am not totally sure and I got this information neither from Table2 nor Figure4 – this information should be added..

I also don't get if the authors these genes for presence/absence on the donkey Y? If they tested, it should be mentioned in the text.

PAB - abbreviation not explained.

Figure1: maybe the assembly gaps should be shown

Figure 6c: I am a little bit confused about the band in males and females in Figure 6C, I assume it is genomic DNA but this is not mentioned in the legend.

Table S10:

These two primers are the same:

PEQ339.4 F4/R4 TTGCCAGGACCACCTCGAATTT TTGCCAGGACCACCTCGAATTT

Reviewer #3 (Remarks to the Author):

Janečka et al. assembled the full male-specific region of the horse Y chromosome and described its main features. Sequencing and assembling the horse MSY is an important achievement, particularly because reference genomes often exclude the Y sex chromosome. However, the

manuscript has a great number of problems, specifically in the analyses and conclusions.

I find the manuscript long and having an increasingly speculative tone. The main conclusions would require further validations and analyses. Furthermore, I strongly advise the authors to perform a careful revision of the scientific literature on the sex chromosomes field and related topics. References are missing, out-dated or simply wrong. The analyses have been performed countless times in previous publications and did not produce considerably new results after including the horse Y chromosome. Besides the full Y chromosome assembly, which represents an incredible amount of work, it was difficult to find something truly novel in the manuscript. Particularly, because some of the authors published in 2011 an article entitled "A Gene Catalogue of the Euchromatic Male-Specific Region of the Horse Y Chromosome: Comparison with Human and Other Mammals" (ref. 19) where they already presented an important number of results and conclusions, including a draft of the horse MSY map based on Y-specific BACs (see Fig.2 in ref. 19), an important number of Y-linked genes (Table 1 in ref. 19), including many of the horse-specific transcripts such as ETSTY1-6; they also indicated in the 2011 publication the copy number and tissue expression profile of the Y genes, and compared the horse MSY with that of the donkey and other mammals, etc. It would be necessary to indicate in the manuscript the differences between both studies and the truly novel questions the authors were able to answer with the fully assembled Y chromosome. For the moment, the authors simply say about the 2011 publication: "The gene content of the horse Y has been examined from sequencing of cDNA libraries, and includes several testis-specific transcripts unique to the equid lineage (ref. 19)", a sentence that I find incomplete given the shared scope and results between the two studies. I'm confident many novel and fascinating things can be found using fully assembled Y chromosomes.

Detailed comments on the manuscript:

1. Lines 39-40 "The eutherian sex chromosomes evolved from a pair of autosomes that diverged around 240-320 million years ago (MYA)". These estimates are based on pair-wise dS calculations from rather old publications (1999-2001). We now know this method is inaccurate. Stochastic variation in divergence estimates, coupled with mutation rate variation among genes, may make age estimates based on synonymous substitution rates ambiguous. If the authors had performed bootstrapping and calculated 95%CI they would have realized that the error rates for many Y-linked genes are larger than 100 MYA (see the work of Kateryna Makova on primate sex-linked genes), which is a clear indicator of the uncertainty and/or variation (stochasticity) in dS estimates. Many authors in more recent publications have preferred phylogenetic-based analyses to estimate the age of sex chromosomes. In fact, there is a Nature article from 2014 where the age of the mammalian sex chromosomes is carefully estimated (see ref. 26) and it dates the origin to 180 MYA, a number that makes more sense in the light of what we know about the three major mammalian clades. For example, if the sex chromosomes had originated around 240-320 MYA, as the authors suggest, it would mean that all three major groups of mammals shared a common sex system (monotremes and therians diverged 200 MYA), a fact that disregards years of scientific work on the sex system of monotreme mammals (platypus and echidna). I strongly encourage the authors to carefully read the work of Jennifer Graves and her colleagues.

2. Lines 103-105. The horse MSY is rich in repeated elements, has a low GC, few unique genes, palindromes, etc. All of these features have been seen in other Y chromosomes. Are there any particular features specific to the horse MSY?

3. Lines 108-109. "the majority of ancestral X-Y and autosomal transposed genes", which ones are missing?

4. Lines 111-112. "almost 100% identical, tandem or disperse repeats" How many are tandem repeats and how many are disperse repeats. Are the frequencies of these elements on the Y chromosome any different from autosomes and the X chromosome?

5. Lines 113-114. "contained genes that were intact in the center, but incomplete towards the ends of the palindrome arms" Are there any particular patterns associated with the incomplete sequences (specific break-points or repeats, for instance)?

6. Lines 122-123. "in the most recent common ancestor of both clades some 4.0-4.5 MYA" To exclude independent acquisitions, which are not uncommon, phylogenetic trees of ARSFY or ARSHY genes would be needed.

7. Lines 127-128 "a novel sequence class containing an array of an equine testis-specific transcript 7 (ETSTY7)". All Y chromosomes studied so far have an array of testis-specific transcripts. The same applies to all autosomes and X chromosomes studied so far. There's nothing truly exciting about having testis-specific transcription (see below for more details) without further data.

8. Lines 148-149. "contrasts with the assumptions made prior to the genome-sequencing era that mammalian MSYs are gene poor and functionally inert". This is not entirely correct. A more accurate analysis would have taken into account the frequencies of repeated genes and the frequencies of unique genes in autosomes and X chromosome. Then one would need to compare these values using a statistical test (probably the Fisher exact test). Besides, many of the repeated genes are likely non-functional. Do the authors detect expression levels in all the copies of the ampliconic genes?

9. Line 151. "X-Y genes" are commonly known as gametologs.

11. Lines 157-159. "Notably, three of these (TAB3Y, SYPY and WWC3Y) have ancient divergence dates (92.6 to 115.4 MYA)". This claim requires further validation and a new set of analyses. For each Y gene, the authors show time trees obtained using the program MEGA. But these trees do not include statistical support (bootstrap values) at key nodes; the trees only show age estimates. One would first need to verify whether key nodes on the trees are statistically supported and only then run the age estimates (preferably using PAML). Age estimates should be shown with 95%CI.

12. Line 162. "The majority of horse X-Y genes were single-copy with broad expression in adult tissues". These results are likely not very precise. Expression patterns were estimated using RT-PCR. One important reason why people commonly use RNAseq comes from the fact that lowly expressed genes are very difficult to quantify with RT-PCR experiments. And given the fact that Y and W-linked genes have generally low expression levels (see Zhou et al PMID: 25504727 and analyses in ref. 26), hence, many of the broadly expressed genes can have tissue-specific expression patterns that would not be detected using RT-PCR. This needs to be changed.

13. Line 162. Based on the results shown in this manuscript it seems that most likely HSFY and TSPY were ampliconic already in the ancestor of all placental mammals.

14. Line 167. HSFY/X, TSPY/X, and CULB4X were already known to have specific-expression in the mature testis in the mouse.

15. Lines 171-174. "resulting in SRY deletion and subsequently leading to disorders of sexual development such as the XY sex reversal syndrome". I wonder what's the rate of SRY microdeletions in the horse? Is it higher than the rate observed in humans? It would be important to add some extra data here.

16. Lines 175-205. "The horse Y genes were aligned with available X homologs to examine phylogenetic patterns and estimate the divergence between the gametologs..." These analyses have been performed countless times in the last decade. And there's already a general agreement about which Y genes are ancient and which ones are lineage-specific. Moreover, defining sex chromosome strata has turned out to be more difficult than initially appreciated, and this is especially so given some evidence that recombination suppression is not necessarily a discrete

process that always occurs in a step-wise fashion. Nevertheless, the authors decided to repeat the same set of analyses (including the horse, donkey, etc. Y genes) that many others did before them, and found, as expected, the same results many others have published before: ancient genes are still ancient (e.g. DDX3Y), polyphyletic gene are still polyphyletic (e.g. AMELY) and lineage-specific gene are still lineage-specific. This section should probably focus only on the horse-specific genes.

17. Lines 184-185. "We assigned X-Y genes to evolutionary strata based on nucleotide divergence of synonymous sites (Ks)". These are old-fashioned analyses scientist performed a decade ago. We now know, as I mentioned above, that stochastic variation in divergence estimates, coupled with mutation rate variation among genes make age estimates based on synonymous substitution rates incredibly ambiguous. Bootstrapping values and 95%CI should be included. These values in other species have shown that the error rates for many Y-linked genes are extremely large, which are clear indicators of the uncertainty and/or variation (stochasticity) in dS estimates. More rigorous methods have been published in recent years.

18. Line 191. The idea of combining stratum 2 and 3 was originally introduced in 2013 by Gang Li, William Murphy et al. (Genome Research), two of the co-authors of the manuscript, not in ref. 12.

19. Lines 237-238. "The function of these Y-born transcripts remain enigmatic, though some, like ETY7, represent novel equine non-coding RNAs and might have regulatory roles" Is there any direct evidence of this regulatory function other than testis-specific expression? Almost the entire genome is expressed in this tissue (see below).

20. Lines 259-260. "No ETSTY7 sequences were detected in the rhinoceros". The analyses shown are insufficient. If ETSTY7 is a long-non-coding RNA, selection acts on these elements to preserve the structure but not the nucleotide sequence, except during short-evolutionary periods. Horse and zebra diverged 4-7 MYA, so FISH analyses could still recover a signal. However, horse and rhinoceros diverged 55 MYA. Given the noncoding nature of ETSTY7, FISH does not seem to be the best approach to detect or discard the presence of this gene in distant species. This needs to be corrected.

21. Lines 273-274. The fact that ETSTY7 could come from the parasite *Parascaris* is an extraordinary claim, and as such, it needs extraordinary evidence. The evidence shown is not convincing enough. The authors need to show, beyond any reasonable doubt, that the libraries sequenced did not contain contaminating *Parascaris* DNA. Intestinal parasites can sometimes be found in the host bloodstream and their DNA could have therefore been mixed with the horse DNA during whole-DNA extraction. Furthermore, parasites often contain genomes with low GC content. It is therefore not unlikely that if a mix of horse/parasite DNA was sequenced, some low-GC-low-complexity sequences from the Y chromosome and the parasite were combined to form chimeric sequences. The fact that one particular region in the horse ETSTY7 gene is similar to *Parascaris* (intron 2/3 and exon 3) is exactly what one could expect if this gene were a chimeric sequence. The authors need to show that the libraries are free of any contaminating *Parascaris* sequences, that the ETSTY7 gene (including intron 2/3 and exon 3) is located in one single DNA molecule, and that the ETSTY7 gene is indeed absent in other mammalian species.

22. Lines 278-279. "A multiple sequence alignment of horse-derived and *Parascaris*-derived ETSTY7 segments was used to reconstruct a phylogenetic tree, which was polyphyletic (Fig. 6)" Nothing can be concluded from the trees shown in Fig6. The trees are unresolved.

23. Lines 284-287. "Altogether, our data provide strong evidence of horizontal transfer (HT) between the horse and its parasite of a putatively functional ampliconic sequence. To the best of our knowledge, this is the first HT described in equids, and among only a few verified HTs within vertebrates." The authors do not show sufficient evidence to support this claim. *Parascaris* DNA

contamination and the presence of chimeric sequences need to be discarded first.

24. Lines 290-291. "The testes-specific transcription of ETSTY7 suggests it functions in male fertility in the horse." Here the authors take a gigantic step from performing superficial analyses to major biological claims. All chromosomes (autosomes and sex chromosomes) have a spurious testis-specific transcription. Deep sequencing of the testis transcriptome of mouse resulted in around 15,000 protein-coding genes expressed in testis and countless numbers of non-coding RNAs and repeated elements. Is it safe to assume that all of these transcripts have a function in fertility and spermatogenesis? No. Testes have a massively noisy expression due to the histone/protamine replacement and the removal of methylation markers, which make almost all the genome accessible to transcription. Therefore, testis-expression does not indicate, by no means, male fertility. I encourage the authors to read the literature regarding the noisy transcription in testis due to the histone/protamine replacement.

25. Lines 324-325. "We hypothesized that MSY genes with comparable expression in horse and donkey testis but significant dysregulation in mules, likely have roles in spermatogenesis." Why would this be the case? I have some difficulty understanding the logic behind this hypothesis. Changes in gene expression in mules compared to donkey and horse may be explained due to general genetic hybrid incompatibility, unrelated to male fertility. The idea of comparing donkey, horse, and mule is fine. But the comparison needs to be performed at a genomic level. That is, how many genes (autosomal and sex-linked) are similar in horse and donkey but different in mules? How many (autosomal and sex-linked) are related to spermatogenesis? Are the three Y genes peculiar (important) cases or are among hundreds of miss-regulated genes in mules?

26. I enjoyed the section about "The Y chromosome diversity among horses", lines 327-364. The analyses felt truly novel and refreshing.

Manuscript NCOMMS-17-32840

Response to reviewers' and editorial comments

We thank the reviewers for a very thorough, insightful, and constructive critique. We found their comments very beneficial to our paper and believe that it has greatly improved our manuscript. We would also like to thank *Nature Communications* for providing us with the opportunity to present a revised manuscript. As follows, please find our responses and list of revisions made in the manuscript. Please let us know if you would like us to provide any further information or clarification.

Note: our responses are in blue color; changes made in manuscript text are shown in red color.

Reviewer #1

Sequencing of mammalian Y chromosomes is a complex task, and one that is essential for our understanding of sex chromosome evolution. Janečka et al have generated the first horse Y chromosome sequence assembly, based on sequencing of ordered BAC clones. Their sequencing strategy was sensible and has given good coverage of both single copy and ampliconic regions of the eMSY, albeit with some caveats as to final clone orientations and positions. About 9.5Mb sequence has been produced from a ~45Mb chromosome, and since the chromosome is ~2/3 heterochromatin this represents a good proportion of the euchromatin.

Gene content of the Y was assessed by RNASeq, with gene structure additionally verified using genomic PCR on males and females, plus RT-PCR on selected tissues. This was valuable to confirm the gene annotation the authors performed, and supports the existing data on horse Y gene expression.

The authors have also carried out phylogenetic and evolutionary analyses of the sequences on the chromosome, most notably finding evidence for horizontal gene transfer from an intestinal parasite. Sufficient information is given to replicate the phylogenetic work, and the analyses are appropriate for this study.

This is a well written and interesting manuscript that fills an important gap in mammalian genomics, and I am happy to recommend it for publication. I have some specific comments below:

Major Comments

Comment 1. The comparative expression between donkey, mule and horse is interesting. The differential regulation of genes based on the genomic background is something that has been seen in mice - see for example the Slx/Sly genomic conflict (PMID: 23028340). These ampliconic genes on the X and Y are competing to distort their transmission ratio, and one way this is mediated includes a broad regulation of sex chromosome expression. A phenotype only emerges when X and Y chromosomes from different backgrounds (with different copy numbers of the relevant genes) encounter each other, and no longer balance each other out. I wonder if a similar situation is involved on the horse sex chromosomes, and if the authors have already considered this?

Response 1. The reviewer makes an excellent point regarding the phenotypic effect of the balance between Y ampliconic genes and their ampliconic counterparts in the X (or autosomes).

This is an appealing and very relevant idea that has the potential to uncover important evidence for genetic conflict. However, the current state of knowledge about ampliconic X-gametologs or autosomal paralogs in the horse is extremely limited because many are missing from the current horse EquCab3 reference genome. Much less is known about such genes in other equids, including the donkey. Certainly, this will be an excellent premise to follow up with as the ampliconic regions of the genome are resolved via technical advancement in sequencing and assembly.

Lines 353-357: Added 2 sentences to the end of the section “Candidate MSY genes for stallion fertility” (page 18):

It is also possible that Y gene regulation and fertility phenotypes in equids depend on the numerical and functional balance between ampliconic gametologs - something similar to how the Sly/Slx loci regulate sex chromosome transmission and male fertility in mice (Cocquet et al. 2012). However, current knowledge about such genes in the horse/equids is too limited even for speculations.

Comment 2. The selected Y genes such as AMELY with low Ks described in stratum S2/3 should be tested for gene conversion, which may help identify the mechanisms at work.

Response 2. We agree that this is an important addition to understanding the mechanisms that have shaped the Y chromosome and affected gene evolution. We tested a total of 16 genes that had lower than expected Ks and recent divergence dates from gametologs or autosomal paralogs using the methods of Sawyer (1989) as implemented in the program GENECONV. Among these genes, our results recapitulated previously published observations regarding gene conversion in AMELY and ZFY (Bellott et al. 2014). In addition, 11 other genes (total 13) tested positive for gene conversion revealing that this process is quite pervasive in the Y.

We now added this information into Figure 4, Table 2, and several places in the manuscript text (lines 189-196; 208-211; 217-218; 221-222) and Methods.

Comment 3. Line 227 - '[the genes] have yet to acquire a testis related function'. I'd rephrase this as 'have not acquired a testis related function' - transposed genes are not guaranteed to gain novel functionality.

Response 3. Changed as suggested.

Comment 4. The ETSTY7 story: were any RT-PCRs carried out on donkey or mule to see if the X copies are expressed in testis as well as Y copies? The RNASeq data in figure S7 looks to show ETSTY7 sequences expressed in horse testis strongly, and a much lower level in donkey. Presumably the donkey transcripts must be coming from X copies?

Response 4. In our 2011 publication (Paria et al. 2011), we showed that *ETSTY7* (alias *ZNF33bY*) was different in horse and donkey: i) PCR amplicons of *ETSTY7* were the same in donkey males and females, and ii) RT-PCR in donkey testis suggested no or very low level expression. At that time, we did not know that these sequences are only in the X chromosome in the donkey. In this manuscript, we briefly mentioned these early observations in Supplementary Information 4.4. **MSY testis transcripts in horse, donkey and mule; Horse-donkey comparison.**

In response and to improve integration of prior and new information in the manuscript, we changed language in:

Lines 271-282

Ampliconic ETSTY7 sequences were also detected in other equids but with different distribution patterns (Fig. 6, Fig. S7). In donkey, the sequences were found in Xq only, suggesting that the low-level transcription of ETSTY7 as previously observed by RT-PCR¹⁹ and by RNAseq in this study, originate from X chromosome copies. In the zebras, ETSTY7 sequences were in Y, Xpq, and the subtelomeres of several autosomes. No ETSTY7 sequences were detected by FISH or PCR in the rhinoceros, an evolutionarily distant Perissodactyl species (SI; Figs. 6; S7; S8). We propose that the ETSTY7 transcript family was acquired by the Equidae X chromosome after divergence from other Perissodactyls about 52-58 MYA³⁰, but before splitting into horses, asses and zebras 4.0-4.5 MYA²³. Subsequently, the sequences were amplified and transposed to the Y chromosome in horses, and to the Y and autosomes in zebras, but not in the donkey (Fig. 6). Alternatively, it may be that ETSTY7 sequences in donkey Y are not amplified enough to be detected by FISH.

b) Supplementary Information 4.4. MSY testis transcripts in horse, donkey and mule Horse-donkey comparison:

As already observed in 2011 (Paria et al. 2011), donkey does not carry male specific ETSTY7 sequences (alias ZNF33b) as indicated by the same PCR amplicon pattern of these sequences in male and female donkeys (Paria et al. 2011). Expression analysis by RT-PCR in donkey testis in the prior study indicated no or very low-level transcription. Here we clarify and refine these observations and show by FISH that ETSTY7 is restricted to the X chromosome in donkeys (see below HT and Fig. S7) with low-level transcription in tests as suggested by RNAseq.

Comment 5. Can the horse X and Y ETSTY7 copies be distinguished at the sequence level? That is, are they evolving in separate directions, which may indicate novel functionality?

Response 5. We agree with the importance of determining how divergent the Y copies are from the X. We therefore performed a more thorough phylogenetic analysis that included 46 additional sequences to address this important question (see revised Figure 6 and Table S6). First, we extracted 14 horse ETSTY7-like sequences using BLAST from the equine genome assembly generated from a female horse; these originated in the X chromosome (8), chromosome 2 (2), and unplaced scaffolds (4). We also extracted ETSTY7-like sequences from the Przewalski's horse (15 sequences, all from unplaced scaffolds) and donkey (18 sequences, all from unplaced scaffolds) assemblies. Further, we PCR-amplified and sequenced ETSTY7 in one male and one female horse (2 sequences), and additional *Parascaris* adults, eggs, and gonads (7 sequences, which matched the previous sequences in our original submission). The 46 additional ETSTY7-like sequence were combined with the 27 sequences we originally analyzed. We aligned them with MUSCLE and reconstructed the phylogenetic tree using maximum likelihood with 1000 bootstrap replicates in RaxML. This comprehensive analysis revealed that the most divergent ETSTY7-like sequences are those from the horse X chromosome and donkey unplaced scaffolds. The ETSTY7 sequences of known equine Y origin (i.e., the ones we sequenced from Y BACs) do form a clade (97% BS value), however, this Y clade also includes 3 equine X copies, in addition to 7 Przewalski's horse sequences, and 6 *Parascaris* sequences. Of these *Parascaris* sequences, three come from the genome assembly in GenBank, while we sequenced other directly from two whole worms (originating from different hosts), worm eggs, worm L4

larva, worm gonads, and worm body wall. The clustering of the Y copies and their divergence from the majority of other horse X copies suggest they may be evolving separately, however, there are a few X copies that do share some similarity. We do think that this may be consistent with novel functionality, however, more in depth analysis and functional experiments need to be performed before presenting such hypothesis and therefore to be cautious we avoid making substantial claims on this hypothesis.

Comment 6. The dot plots in Figure S4 suggest two diverging patterns to the repeat when comparing the 50bp and 100bp resolution; there are distinct blocks visible at 100bp which are otherwise similar at 50bp resolution. Do the underlying sequences give an indication of what has happened? For example, could this reflect intrachromosomal inversions mixing diverging sequences?

Response 6. The dot plots in Fig. S4 reflect the level of identity the regions have with one another. There does not appear to be divergence in repeat pattern when comparing the different block sizes (50, 100, 500, 1000 bp). There is, however, a differing level of identity between regions when the length of the sequence compared increases. We do not hypothesize a complex mechanism for this decrease, rather it is simply reflective of the accumulation of mutations since the expansion of these regions, and therefore less identity.

As a response, we added a sentence to the legend of Figure S4.

Figure S4: Intra-chromosomal sequence similarity. *Triangular dot plots showing the location of 100% identical sequences in horse MSY using sequence motifs (word size) of 50 bp, 100 bp, 500 bp and 1000 bp with a step of 20% of the word size. Note the accumulation of 100% identical sequences specifically in the major ampliconic region at eMSY ~ 1-4 Mb (seen as a pyramid shape). The decrease in identity observed as word size increases is reflective of the increase in divergence between homologous sequence blocks since the expansion of this ampliconic region.*

Comment 7. Within the supplementary tables S6, pig gene accessions refer to the Vega website; since pig genome 11.1 went live, the Vega gene accessions have been archived. It would be helpful to update the pig accessions to their live Ensembl IDs.

Response 7. Replaced Vega accessions with live Ensembl IDs for all, except *AP1S2Y*, *USP9Y* and *ZRSR2Y* because no stable ID for these genes is present in Ensembl.

Comment 8. The autosomal transposed genes have a broad expression pattern. Is this also true of their autosomal paralogues?

Response 8. A brief answer is that we do not know, because only limited expression data are available for the horse genome (Ensembl: cerebellum, embryo, placental villosus and testis). Functional annotation of the horse genome is ongoing (FAANG project) but yet unpublished and not available at NCBI or other genome browsers. However, personal communication with Dr. Carrie Finno, one of the leaders of the horse FAANG initiative, gave us access to the expression data for some genes of interest. The data is based on 34 tissues (from 2 female horses), including seven somatic tissues (brain, kidney, heart, muscle, liver, lung, spleen) that were used for RT-PCR in this manuscript (Figure S5). Expression profiles of autosomal paralogues of 6 genes (*ATP6V06*, *EIF3C*, *HSPA1L*, *HTRA3*, *RPS3A*, *SH3TC1*) are broad; *MYL9* is expressed in 9 out of 34 tissues and only in lung from our panel; no expression data are available for *OR8J2*,

OR8K3 and *TIG1* because the two olfactory receptors are not well annotated and *TIG1* is a transposon found at multiple locations in the horse genome.

Because the expression data is incomplete and unpublished, we only slightly rephrased the text.

Lines 238-241: *Their broad expression pattern, similar to their autosomal paralogs (C. Finno and E. Burns, personal communication), is noteworthy because other multi-copy Y genes are exclusively or predominantly expressed in testes. It is possible that because the recent origin, they have not yet acquired a testes related function.*

Comment 9. Figure 4 is quite hard to interpret. It is difficult to track the stratum versus age colouring in panel A especially. Please try redrawing this figure; perhaps changing the bars to points, with a symbol representing the type of gene would be clearer.

Response 9. We have revised the figure as follows: i) in response to comments by Reviewer #3, we reconstructed gene phylogenies with bootstrap replicates and recalculated X-Y genes' divergence times. The new data did not differ essentially from what we had before, though there were minor changes in divergence times. The A-portion of the figure has been changed accordingly, and ii) based on phylogenetic trees, all genes fall into 2 main categories – monophyletic and polyphyletic. Thus, as requested by the reviewer, we simplified color-coding to 2 colors in the A-portion of Figure 4.

Reviewer #2

In the paper "Horse Y chromosome assembly displays unique evolutionary features, putative stallion fertility genes and horizontal transfer", Dr Raudsepp and co-workers present a first almost complete assembly of the male specific regions of the horse Y-chromosome. I really appreciate the work the authors performed to provide this comprehensive analysis. They not only show functional annotation, in addition they did multiple analyses to address very interesting aspects. This ranges from gene content on the horse Y, over prediction of horse MSY genes responsible for stallion fertility to evolutionary dynamics and lineage specific. This manuscript provides a comprehensive view on this particular part of the horse genome. It covers a very interesting topic which will likely gain great interest in the field.

The horse MSY assembly is highly awaited in the horse community and due to the widespread analysis performed, the manuscript will attract interest from geneticist and breeders also from other animal genome communities.

The manuscript is clearly written (some minor comments are pointed out below). The Authors provide detailed methodological information so that the experiments could be understood. Most claims are sound to me and appropriately discussed with regard to previous literature.

Overall the paper is a high quality study and I recommend the manuscript for publication except one aspect needs an additional test to improve reliability at this state in my opinion.

Major Comment: Coming to my major point: Horizontal Gene Transfer from *Parascaris* to the eMSY. The authors first describe a horse Y-chromosomal ETSYT7 multi copy element. As the ETSYT7 element is present also in donkeys and zebras but not in the Rhino it must have its origin in the ancestor of the Equine lineage prior to the split of horses from donkeys/zebras about 4-5 million years ago. In horses ETSYT7 is detected on the eMSY and on the X and whereas it is

found also on other chromosomes in other equids indicating mobility of the motif in the genome. After observing homologous ETSYT7 sequences in an assembly of the horse intestinal parasite 'Parascaris equorum' the authors hypothesize that the ETSYT7 sequence came into horses from the Parascaris genome via horizontal gene transfer.

This is a somehow unprecedented observation and the authors performed experiments to prove their hypothesis (Figure 6). But I am not totally convinced that their experimental tests rule out a possible contamination of the worm DNA with horse that would refute their assumptions.

- Regarding the Parascaris assembly: when I Blast the horse mtDNA sequence (NC_001640) to the Parascaris assembly (GCA_000951375.1), I find a perfect match of the horse mtDNA with 'Parascaris >LM463828.1 Parascaris equorum genome assembly P_equorum, scaffold PEQ_contig0000001'. This means to me, that the Parascaris assembly is contaminated with horse sequences. The Parascaris assembly could therefore also be contaminated with horse ETSY7 elements and to my opinion it should be proven carefully, if the ETSY7 element is really embedded in the Parascaris genome.
- The authors tested the occurrence of the ETSTY7 repeat in the Parascaris genome by PCR amplification of ETSTY7 using genomic DNA isolated from different worm tissues and worm developmental stages as template. The worms were collected from affected horses. In Figure 6A.1 the Parascaris control amplifies similar in all samples, whereas the ETSTY7 signal does not.
- I think that the experiments shown in the manuscript cannot definitely confirm, that the PCR product shown for ETSTY7 (Figure 6 A, 3) derives from the worm genome and not from horse contamination.
- The authors did only one experiment to test that ETSTY7 derives from the worm and not from horse DNA contamination. As a prove they use the absence of the horse Y-chromosome specific TSPY signal in the worm tissues (Figure 6A, 2). For my opinion TSPY is not an ideal control for horse DNA contamination in this context because a) TSPY is male specific. In case of the worms were extracted from an affected female, TSPY would not amplify even in case of horse DNA contamination. b) ETSTY7 is occurring in multiple copies on the X and the Y of the horse. Therefore PCR amplification is probably much more efficient for ETSTY7 than for the 13 copy TSPY gene (ETSYT7/exon3 primers give a much more prominent band than TSPY Fig6A).

My suggestion is to test the absence of horse DNA with a more appropriate (or several) horse DNA amplification controls. One could use horse multicopy autosomal loci or mtDNA instead of TSPY, to convincingly show that the ETSYT7 signal does not derive from horse contamination in the worm sample.

Also it would be good to see these tests (because of sensitivity, amplification efficiency, copy number estimate) in a qPCR experiment.

Response to Major Comment. We absolutely agree with the reviewer that more and better-designed experiments are needed to test for horse-to-parasite DNA contamination and validate or refute the proposed horizontal transfer (HT) event. Thank you for suggesting additional approaches.

In response, we did the following:

1) Isolated DNA from additional parasite samples originating from different hosts, developmental stages (including eggs) and treatment groups. These include DNA from an adult worm (A3) that was incubated 48 h *in vitro* (outside the host) before subject to DNA isolation. Details of all *Parascaris* samples used for PCR tests are presented in **Supplementary Table S16**. Our rationale is that in case of contamination, we should see inconsistencies in PCR patterns across different worm DNA samples. On the other hand, if it is true HT or no HT, PCR results should be consistent across all samples, in one way or another.

2) Tested possible horse-to-parasite contamination using multiple horse MtDNA primers (positioned to target regions across the mitochondrial genome) and primers for known and published autosomal CNVs (such as olfactory receptors; Ghosh et al. 2014). Briefly: we did not find evidence for horse-to-parasite DNA contamination.

The results of additional testing are shown in essentially revised **Figure 6** and a new Supplementary **Figure S8**. Manuscript text in the section “**Massive amplification and horizontal transfer of a testis transcript ETSTY7 in horse and equids**” (lines 259-323) has been revised.

3) Carried out additional experiments to test for possible parasite-to-horse contamination (concern of Reviewer #3). We used PCR with parasite-specific primers on horse gDNA and on DNA from multiple BAC clones from two different BAC libraries. The results are shown in Fig. 6 and Supplementary Fig. S8, and provide no evidence for parasite-to-horse DNA contamination. Also, it must be mentioned that we routinely use the 612 bp *ETSTY7* PCR product as a FISH probe in clinical cytogenetics for quick visualization of horse sex chromosomes. We see *ETSTY7* sequences in all horses (dozens tested) and consider this as a strong evidence that these sequences are a legitimate part of the horse genome and not a result of contamination with parasite DNA. Furthermore, the majority of horses are treated with anthelmintics and if untreated, *Parascaris* typically affects only very young horses (yearlings). For example, parasite samples for this study originate from a special untreated study herd in Kentucky.

4) Essentially revised Supplementary Information *Chapter 5. Horizontal transfer; 5.1. Discovery and validation*; included new Figure S8 to Supplementary Information and essentially revised Figure 6 in the manuscript.

5) Finally, the reviewer was concerned about uneven PCR amplification of *ETSTY7* from worm DNA templates, compared to the amplification with *Parascaris*-specific primers. This is due to the following reasons:

DNA quality. *Parascaris* DNA quality was consistently inferior to that isolated from horses and did not perform equally well in PCR, particularly for amplifying large PCR products. We saw some improvement after *Parascaris* DNA additional clean-up through Qiagen columns (DNeasy Blood & Tissue kit). Nevertheless, the parasite DNA remained fragmented and was not an ideal template for amplifying >600 bp PCR products. In our original Figure 6, the parasite-specific primers amplify a 214 bp product, while *ETSTY7*-3ex3 product is 612 bp. In the revised Figure 6, we included *ETSTY7*-PCR on additional and better *Parascaris* DNA templates with more even amplification. Most importantly and in support of the proposed HT, inclusion of additional *Parascaris* samples and improving DNA template quality, amplification of putative horse-parasite HT sequences improved and was more consistent;

We used heterologous primers. It is important to underline, that ETSTY7-3ex3 primers are heterologous – designed from horse eMSY sequence, showing 3 mismatches (F-primer 2 mismatches; R-primer 1 mismatch) with *Parascaris* contig0000339. Notably, if we designed primers from another region of ETSTY7, or conversely, from other contigs of the *Parascaris* genome, no cross-horse-parasite amplification was obtained. Also, if we increased PCR stringency for ETSTY7-3ex3 primers, we continued amplifying from horse gDNA but lost *Parascaris* products. Our rationale is that if there was a true horse-to-parasite contamination, PCR conditions should not matter so much;

In response to the suggestion to study ETSTY7 in Parascaris by qPCR. First, we think that the uneven amplification from parasite templates is mainly a combined result of suboptimal parasite DNA quality and primer mismatches. Secondly, the ETSTY7-3ex3 primers amplify a 612 bp product and are, thus, not suitable for qPCR. We tried to design from the same region a primer set with a smaller (<200 bp) product but these primers had 5 mismatches with *Parascaris* and did not amplify at all. As a result, we did not conduct qPCR experiments.

#Minor comments:

Unfortunately no pages, I therefore refer to the sections:

1. Section-Unique features of the horse Y chromosome

L103-111: Over half of eMSY is repeats..... in L108 they write over 60% of the eMSY consists of single copy sequences. This is confusing.

Thank you, it certainly is. Changed as follows (lines 106-109) :

In the remaining non-repetitive eMSY, we identified 4 distinct sequence classes: single-copy (~60%), multi-copy/ampliconic (~37%), PAR transposed and novel XY ampliconic array (Fig. 1B). Single-copy sequences contained the majority of ancestral X-Y (25 of 29) and autosomal transposed (7 of 10) genes.

2. Section- autosomal transposed genes

a) L215:with three very recent events occurring after the divergence of horse and donkey. Which genes do you refer to here? I assume it is EIF3CY, MYL9Y and SH3TC1Y? But I am not totally sure and I got this information neither from Table2 nor Figure4 – this information should be added.

Response. Per request of another reviewer, we improved divergence time analysis using MCMCTREE with the independent rates model and soft constraints. There are 9 genes that date close to the 4.0-4.5 MYA divergence of horse and donkey: NLGN4Y, OFD1Y, TMSB4Y, ARSFY, ARSHY, EIF3CY, HTRA3Y, MYL9Y, and SH3TCY. Corresponding changes have been made in **Figure 4** and **Table 2**. We also added **the confidence intervals** to Table 2. Improved timetrees are presented as **Figure S9** in Supplementary Information and the Ks values are in **Table S15**. The text of the section ***The X-Y genes (gametologs) and evolutionary patterns*** has been essentially revised (lines 157-224).

b) I also don't get if the authors these genes for presence/absence on the donkey Y? If they tested, it should be mentioned in the text.

Response. We compared the gene content of horse and donkey Y chromosomes in a prior study (Paria et al. 2011) and showed that *EIF3CY* was not found in the donkey. In response to the reviewer's query, we conducted regular qualitative PCR with horse *MYL9Y* and *SH3TC1Y* primers in donkey, which amplified in both males and females. These results are inconclusive, meaning either that these genes are not present in donkey Y or that Y and autosomal paralogs are indistinguishable by our PCR as designed. Testis RNAseq shows very low abundance of *SH3TC1Y* transcripts in donkey compared to horse. This does not inform on the Y-linked nature of this gene in the donkey, only to its testis expression. We have no clear indication, one way or another, for *MYL9Y*. No changes made because of limited data.

3. PAB - abbreviation not explained.

Corrected.

4. Figure1: maybe the assembly gaps should be shown.

Locations of the three gaps in the BAC contig map (Fig. S1) are now shown in the MSY sequence and gene map Figure 1.

5. Figure 6c: I am a little bit confused about the band in males and females in Figure 6C, I assume it is genomic DNA but this is not mentioned in the legend.

Thank you. Corrected (though we essentially rearranged this figure including the legend).

6. Table S10:

These two primers are the same:

PEQ339.4 F4/R4 TTGCCAGGACCACCTCGAATT TTGCCAGGACCACCTCGAATT

Corrected.

Reviewer #3

I find the manuscript long and having an increasingly speculative tone. The main conclusions would require further validations and analyses. Furthermore, I strongly advise the authors to perform a careful revision of the scientific literature on the sex chromosomes field and related topics. References are missing, out-dated or simply wrong. The analyses have been performed countless times in previous publications and did not produce considerably new results after including the horse Y chromosome. Besides the full Y chromosome assembly, which represents an incredible amount of work, it was difficult to find something truly novel in the manuscript. Particularly, because some of the authors published in 2011 an article entitled "A Gene Catalogue of the Euchromatic Male-Specific Region of the Horse Y Chromosome: Comparison with Human and Other Mammals" (ref. 19) where they already presented an important number of results and conclusions, including a draft of the horse MSY map based on Y-specific BACs (see Fig.2 in ref. 19), an important number of Y-linked genes (Table 1 in ref. 19), including many of the horse-specific transcripts such as ETSTY1-6; they also indicated in the 2011 publication the copy number and tissue expression profile of the Y genes, and compared the horse MSY with that of the donkey and other mammals, etc. It would be necessary to indicate in the manuscript the differences between both studies and the truly novel questions the authors were able to answer with the fully assembled Y chromosome. For the moment, the authors

simply say about the 2011 publication: “The gene content of the horse Y has been examined from sequencing of cDNA libraries, and includes several testis-specific transcripts unique to the equid lineage (ref. 19)”, a sentence that I find incomplete given the shared scope and results between the two studies. I’m confident many novel and fascinating things can be found using fully assembled Y chromosomes.

Response to the general comment:

Thank you for drawing our attention to the fact that we had not adequately highlighted the unique features of the horse Y. With our responses to the critique by this and the other two Reviewers, we hope to have underscored more the novel features of horse Y compared to the other sequenced mammalian Ys.

1) In response to the comment that “ *these analyses have been done innumerable times*”, we respectfully argue that: i) with complete Y assemblies available for just 4 species and partial assemblies for 8 species, altogether representing 4 eutherian orders and marsupials, there has not been enough data available for ‘innumerable’ analyses; ii) to our best understanding, there are just a handful of recent and comprehensive comparative analyses of Y genes across mammals, *viz.*, Cortez et al. 2014, Bellott et al. 2014 and Li et al. 2013. Since then, data has become available for pig Y, and with this manuscript, for horse/equids Y, and iii) in one of these comprehensive analyses (Cortez et al. 2014), the authors write: “*However, our understanding of mammalian Y chromosome evolution remains limited owing to the restricted amount and phylogenetic representation of available Y chromosome data.*”

In the present study we “*increase the phylogenetic representation of available Y data*” by characterizing the horse and equids Y. Indeed, and as expected, there are similarities with already published Ys and this is important in understanding the general trends in Y evolution. Furthermore, there are also species and group specific features, which are always novel, such as for example the PAR transposition, record number of gametologs and horizontal transfer as observed in horse Y. Thus, we consider the data presented in the manuscript both novel and important with regards the general and unique patterns of Y evolution. We also think that our analyses indicate likely Y sequences with importance in male fertility in horse and equids. In response to the Reviewer’s general comment, we have reshaped the manuscript and changed language to improve the visibility of novel and unique features of the horse Y chromosome.

2) In response to the comment that most of horse MSY genes and their expression profiles were published in 2011, we argue that:

- We identified 19 new genes not reported in 2011. These genes are noted as ‘new’ in Supplementary **Table S5** and we added this number now into the main text (line 145).
- The Y map in 2011 was tentative and schematic; no information was provided for individual BACs or the actual BAC tiling path map of MSY (**SI, Figure S1, Tables S1, S2**);
- Gene sequences from 2011 were partial transcript sequences as identified by cDNA capture. Here we present gene models in a sequence map;
- In 2011, gene copy numbers were tentatively estimated by cDNA- and BAC-FISH. Here we present sequence-based copy number estimation, which is much more accurate (**Table 2; Table S5**);

- In 2011 gene expression profiles were determined qualitatively by reverse transcriptase PCR on a panel of adult tissues. Here we add RT-PCR data on the same tissues for all new (19) genes. Importantly, with testis RNAseq data we improved gene models, facilitated discovery of additional genes, and allowed quantitative analysis between different genes in horse MSY, and between the same genes across horse, donkey and mule.

In brief, we strongly believe that the present manuscript provides essentially new data and takes structural, functional, evolutionary, and comparative analysis of the horse Y chromosome to a qualitatively new level.

Detailed comments on the manuscript:

1. Lines 39-40 “The eutherian sex chromosomes evolved from a pair of autosomes that diverged around 240-320 million years ago (MYA)”. These estimates are based on pair-wise dS calculations from rather old publications (1999-2001). We now know this method is inaccurate. Stochastic variation in divergence estimates, coupled with mutation rate variation among genes, may make age estimates based on synonymous substitution rates ambiguous. If the authors had performed bootstrapping and calculated 95%CI they would have realized that the error rates for many Y-linked genes are larger than 100 MYA (see the work of Kateryna Makova on primate sex-linked genes), which is a clear indicator of the uncertainty and/or variation (stochasticity) in dS estimates. Many authors in more recent publications have preferred phylogenetic-based analyses to estimate the age of sex chromosomes. In fact, there is a Nature article from 2014 where the age of the mammalian sex chromosomes is carefully estimated (see ref. 26) and it dates the origin to 180 MYA, a number that makes more sense in the light of what we know about the three major mammalian clades. For example, if the sex chromosomes had originated around 240-320 MYA, as the authors suggest, it would mean that all three major groups of mammals shared a common sex system (monotremes and therians diverged 200 MYA), a fact that disregards years of scientific work on the sex system of monotreme mammals (platypus and echidna). I strongly encourage the authors to carefully read the work of Jennifer Graves and her colleagues.

Response 1. We agree and have corrected eutherian sex chromosome origin to 180 MYA and replaced references with *Cortez et al. 2014*.

2. Lines 103-105. The horse MSY is rich in repeated elements, has a low GC, few unique genes, palindromes, etc. All of these features have been seen in other Y chromosomes. Are there any particular features specific to the horse MSY?

Response 2. In response, we have revised the section “*Unique features of the horse Y chromosome....*” to make unique features more visible.

3. Lines 108-109. “the majority of ancestral X-Y and autosomal transposed genes”, which ones are missing?

Response 3. We changed the language and added reference to Table 1. The text reads now as follows:

Lines 108-110: *Single-copy sequences contained the majority of ancestral X-Y (25 of 29) and autosomal transposed (7 of 10) genes. Nearly all multi-copy/ampliconic sequences localized between eMSY:1,000,000-4,700,000 bp (Fig. 1B; Table 1).*

4. Lines 111-112. “almost 100% identical, tandem or disperse repeats” How many are tandem repeats and how many are disperse repeats. Are the frequencies of these elements on the Y chromosome any different from autosomes and the X chromosome?

Response 4. Because the assembly of the Y ampliconic region is tentative, detailed inventory of different types of repeats was premature. We did this only for interspersed repeats identified by RepeatMasker and the data are presented in Table S4. Also, given the state of the current genome assembly (EquCab3), accurately estimating the frequency of repetitive elements between highly repetitive chromosomes (X and Y) is not feasible because many repeated sequences are still misassembled, unplaced, and collapsed. We agree that resolution of repetitive regions in the X should be addressed, but is out of the scope of this work.

In response, we changed language on line 111: *Characteristic to these regions was the presence of high identity repeats (Fig. 1C; Fig. S4).*

5. Lines 113-114. “contained genes that were intact in the center, but incomplete □ towards the ends of the palindrome arms” Are there any particular patterns associated with the incomplete sequences (specific break-points or repeats, for instance)?

Response 5. This is a very good point but it is something that we believe requires additional analysis that we plan to do in a paper that focuses on the multi-copy region. We removed this statement to just say that “multicopy genes included copies with intact and incomplete copies, with missing exons.

Lines 112-113: *Within these, we observed regions which contained both intact copies of genes and those with an incomplete numbers of exons.*

6. Lines 122-123. “in the most recent common ancestor of both clades some 4.0-4.5 MYA” To exclude independent acquisitions, which are not uncommon, phylogenetic trees of ARSFY or ARSHY genes would be needed.

Response 6. We agree with the reviewer and unfortunately in our original submission we did not explain our methods and rationale sufficiently for this section. We did reconstruct maximum likelihood trees prior to the divergence time analysis for all genes. To clarify, we now included both the maximum likelihood trees and divergence time trees into the supplementary file (**Figures S9 and S10**). Regarding the two genes mentioned above, in the updated MCMCTREE analysis ARSHY was dated to 4.3 MYA (95% CI of 4.0 – 4.5) and the ARSFY gene to 2.3 MYA (95% CI of 1.3-1.4). In response to Reviewer 1, in our revision we also performed gene conversion analysis in GENECONV: ARSFY tested positive for gene conversion whereas ARSHY tested negative for gene conversion. These two genes, ARSFY and ARSHY are part of a single contiguous ~200kb block of sequence that aligns with the ARSF/ARSH region of the PAR. We provide independent evidence in the manuscript that this same ARSFY/ARSHY transposition is present in other equids. Therefore, taken as a whole we conclude the most parsimonious explanation is that there was a single PAR transposition of a segment that included both genes into the eMSY prior the divergence of *Equus* (~4.5 MYA), and subsequently there

was a gene conversion event in ARSFY which reduced its age to 2.3 MYA. We clarify our rationale in the text on lines 213-218.

7. Lines 127-128 “a novel sequence class containing an array of an equine testis-specific transcript 7 (ETSTY7)”. All Y chromosomes studied so far have an array of testis-specific transcripts. The same applies to all autosomes and X chromosomes studied so far. There’s nothing truly exciting about having testis-specific transcription (see below for more details) without further data.

Response 7. Indeed, all sequenced Y chromosomes (4 completed and 8 partial) show the presence of such arrays. Nevertheless, it was important to report their presence in horse Y since we are describing the Y of a new and evolutionarily distinct eutherian group. Our focus in this section, however, was not on testis-specific transcription but rather on species-specific features of these ampliconic arrays, particularly the unique features of *ETSTY7* arrays in the horse. We agree that this was not presented clear enough and in response, we revised the text as follows:

Lines 127-130: *Such arrays are characteristic to all Y chromosomes studied so far and thought to be needed to stabilize Y gene content and protect spermatogenesis genes (Bellott et al. 2014, Cortez et al. 2014). However, ampliconic arrays are not conserved and show unique, species-specific features of origin, distribution and sequence properties.*

With regards the Reviewer’s comment that such testis-specific ampliconic arrays are present in all autosomes and X chromosomes, we have very limited information for the horse genome. To our best knowledge, X-linked and autosomal arrays have been described in human and mouse, but not for other eutherian species because in current draft assemblies these sequences collapse and are missing from assemblies. This limits our ability to accurately assess this question, and will be the topic of future study as these regions are more resolved.

8. Lines 148-149. “contrasts with the assumptions made prior to the genome-sequencing era that mammalian MSYs are gene poor and functionally inert”. This is not entirely correct. A more accurate analysis would have taken into account the frequencies of repeated genes and the frequencies of unique genes in autosomes and X chromosome. Then one would need to compare these values using a statistical test (probably the Fisher exact test). Besides, many of the repeated genes are likely non-functional. Do the authors detect expression levels in all the copies of the ampliconic genes?

Response 8. At the present state of horse reference assembly, we intentionally limited comparisons to total gene density and did not attempt to delineate between repetitive / unique genes due to the mis-assembly, collapsing, or exclusion of repetitive sequences in the reference genome, particularly on the X. Such a comparison would be interesting once the repetitive regions of the genome have been resolved, but is out of the scope of this work.

When investigating the expression of ampliconic genes via either reverse-transcriptase (RT) PCR or high-throughput sequencing, in most cases it is not possible to differentiate between the expression of one or another gene copy due to high sequence identity. Both RNAseq and RT-PCR data demonstrate that most genes are functionally active, including ampliconic genes on an exon-by-exon basis (Fig S5, Table S7).

No changes made.

9. Line 151. “X-Y genes” are commonly known as gametologs.

Response 9. This is correct and we use the term at several places in the manuscript. However, we understand that the Reviewer advises to make this clear from the beginning that ‘X-Y genes’ and ‘gametologs’ are synonyms. In response, we added the latter in the section heading on line 157: *The X-Y genes (gametologs) and evolutionary patterns.*

11. Lines 157-159. “Notably, three of these (TAB3Y, SYPY and WWC3Y) have ancient divergence dates (92.6 to 115.4 MYA)”. This claim requires further validation and a new set of analyses. For each Y gene, the authors show time trees obtained using the program MEGA. But these trees do not include statistical support (bootstrap values) at key nodes; the trees only show age estimates. One would first need to verify whether key nodes on the trees are statistically supported and only then run the age estimates (preferably using PAML). Age estimates should be shown with 95%CI.

Response 11. We agree that all methods were not clearly described in the initial submission. As mentioned in **Response 6**, we did perform maximum likelihood phylogenetic reconstruction of all genes prior to the divergence time analysis. Now, we also performed additional analysis in RaxML to validate our previous conclusions. All maximum likelihood trees with the bootstrap values from 1000 replicates are included in the **Supplementary Information as Figure S10**. In addition, we re-analyzed the divergence times with MCMCTREE in PAML using the independent rates model and soft constraints (**Figure S9**). We updated the text, all figures and tables with the revised estimates and added data for confidence intervals (CI) to **Tables 2 and S15**.

Revisions in the text on lines 158-165: *Notably, two of these, TAB3Y and SYPY, have ancient divergence times from their gametologs (123.5 MYA and 115.7 MYA, respectively) and are not Y-linked in any other species studied thus far, indicating that the horse MSY has retained a unique set of ancestral genes (Fig. 4; Table 2). The WWC3Y gene is also unique to horse MSY, but its intermediate divergence time (56.9 MYA) and position near the PAR suggest a more recent expansion of the MSY. In addition, because of a relatively short PAR in the horse, typical mammalian PAR genes, such as ZBED1, TBL1, SHROOM2, and STS^{22, 26}, belong to eMSY in the horse.*

12. Line 162. “The majority of horse X-Y genes were single-copy with broad expression in adult tissues”. These results are likely not very precise. Expression patterns were estimated using RT-PCR. One important reason why people commonly use RNAseq comes from the fact that lowly expressed genes are very difficult to quantify with RT-PCR experiments. And given the fact that Y and W-linked genes have generally low expression levels (see Zhou et al PMID: 25504727 and analyses in ref. 26), hence, many of the broadly expressed genes can have tissue-specific expression patterns that would not be detected using RT-PCR. This needs to be changed.

Response 12. We used reverse-transcriptase PCR, which is a qualitative method and not real-time PCR, which is a quantitative approach. We noticed that for some genes, reverse-transcriptase PCR was more sensitive than RNAseq. These observations are presented and discussed in Supplementary Information (4.3. Discrepancies between RT-PCR and testis RNAseq). Otherwise, we agree with the reviewer that the results are not very precise, though due to different reasons than mentioned by the reviewer. First, reverse-transcriptase PCR was

conducted on 9 adult tissues, which certainly do not represent the entire richness of tissues and cell types in mammalian body. Secondly, we could compare RT-PCR with RNAseq only for testis. Nevertheless, we are confident that the expression data are representative to eMSY genes and comparable with prior publications for other mammals.

The recently started horse genome functional annotation (FAANG) project has generated RNAseq data for female tissues only. We anticipate to essentially improve the functional annotation of eMSY once RNAseq data for male tissues becomes available. Unfortunately, we are unable to do so at this time due to data availability. No changes made.

13. Line 162. Based on the results shown in this manuscript it seems that most likely HSFY and TSPY were ampliconic already in the ancestor of all placental mammals.

Response 13. Thank you, this is an interesting suggestion. However, we think that it is unlikely that *TSPY* was ampliconic in placental ancestor because it is a single-copy in rat and a single-copy pseudogene in mouse. We rather think that it was amplified multiple times. Evolutionary history of *HSFY* seems to be more complicated. The gene was amplified with two to four copies on the X chromosome before the divergence of placental and marsupial mammals, so all therian Y chromosomes had multiple *HSFYs* to begin.

In response, we did not want to add more speculations (the reviewer correctly pointed out that we already have too many) and no changes were made.

14. Line 167. HSFY/X, TSPY/X, and CULB4X were already known to have specific-expression in the mature testis in the mouse.

Response 14. We do not completely understand this comment, particularly because *Tspy* is a nonfunctional pseudogene in mouse and not expressed in mature testis. Further, mouse Y has no gametolog for *CULB4X*, thus testis expression of the X gametolog is irrelevant. No changes made.

15. Lines 171-174. “resulting in SRY deletion and subsequently leading to disorders of sexual development such as the XY sex reversal syndrome”. I wonder what’s the rate of SRY microdeletions in the horse? Is it higher than the rate observed in humans? It would be important to add some extra data here.

Response 15. We thank the reviewer for making this good point. Based on a limited number of publications and our unpublished observations, XY male-to-female sex reversal counts for about 12-30% of all cytogenetic abnormalities in horses, while 70% of XY female horses have lost *SRY*. In contrast, only 10-20% of human XY females (Swyer syndrome) have *SRY* mutations/deletions and 80-90% carry normal *SRY*. Thus, there are clear differences between species and we think that these differences are due to the location, copy number and neighboring sequence environment of *SRY* in MSY. We first noted these differences and discussed their likely association with the molecular organization of MSY in different species in Raudsepp et al. 2010. In response, we briefly expand these observations in the manuscript as follows:

Lines 178-183: *Notably, XY male-to-female sex reversal counts for about 12-30% of all cytogenetic abnormalities in horses, whereas 70% of XY female horses have lost SRY²⁴. In contrast, only 10-20% of human XY females (Swyer syndrome) have SRY mutations and the*

majority carry normal SRY, while the condition in other mammalian species studied is rare or absent^{see 24}. Thus, there are clear differences between species and we propose that this is because of differences in MSY organization and, particularly, the location of SRY.

16. Lines 175-205. “The horse Y genes were aligned with available X homologs to examine phylogenetic patterns and estimate the divergence between the gametologs...” These analyses have been performed countless times in the last decade. And there’s already a general agreement about which Y genes are ancient and which ones are lineage-specific. Moreover, defining sex chromosome strata has turned out to be more difficult than initially appreciated, and this is especially so given some evidence that recombination suppression is not necessarily a discrete process that always occurs in a step-wise fashion. Nevertheless, the authors decided to repeat the same set of analyses (including the horse, donkey, etc. Y genes) that many others did before them, and found, as expected, the same results many others have published before: ancient genes are still ancient (e.g. DDX3Y), polyphyletic gene are still polyphyletic (e.g. AMELY) and lineage-specific gene are still lineage-specific. This section should probably focus only on the horse-specific genes.

Response 16. Although there have been several studies that have performed similar analyses of Y genes, these have not included a comprehensive phylogenetic-based analysis of Perissodactyla and Y gene sequence data integrated with a detailed Y map and functional analysis. Our higher-level phylogenetic approach provides additional insight into the evolution of Y genes and builds a foundation from which we then explore the unique features present in horses. Our analysis revealed several novel insights. First, we provide divergence time estimate of all MSY genes, while in the papers mentioned by the reviewer, the timing of acquisition was not directly estimated. Further, we were able to identify and quantify specific recent gene acquisition events in the horse. There are 16 MSY genes in the horse that have lower Ks and reduced divergence times. We found evidence that 13/16 of these were subject of gene conversion and although there has been previous analysis of gene conversion, it was not done with such a broad phylogenetic sampling. In addition, we have evidence about recent acquisition of 9 genes that date to around the Equus divergence (4-4.5 MYA). This in itself is novel as no one to date has provided a similar analysis.

17. Lines 184-185. “We assigned X-Y genes to evolutionary strata based on nucleotide divergence of synonymous sites (Ks)”. These are old-fashioned analyses scientist performed a decade ago. We now know, as I mentioned above, that stochastic variation in divergence estimates, coupled with mutation rate variation among genes make age estimates based on synonymous substitution rates incredibly ambiguous. Bootstrapping values and 95%CI should be included. These values in other species have shown that the error rates for many Y-linked genes are extremely large, which are clear indicators of the uncertainty and/or variation (stochasticity) in dS estimates. More rigorous methods have been published in recent years.

Response 17. This is indeed an older approach and that is why we did not solely rely on Ks to estimate when the Y and X gametologs diverged, but instead performed a thorough maximum likelihood reconstruction in RAxML followed by divergence time analysis in MCMCTREE. We include the bootstrap support values for the trees in the **Supplementary Information Figure S10** and the 95% CI intervals in Table 2 and Table S15. The reason for showing Ks values is to make our results more directly comparable to the older publications that grouped genes into strata

based on Ks. In our manuscript, we show these Ks values solely for illustrative purposes and do not categorize or estimate origin based on Ks. For example, the Ks of genes across the horse X shows that divergence is lower as you get closer to the PAB, consistent with the previously proposed step-wise evolution of the MSY via inversions that block recombination. This pattern seen in the horse X contrasts the rampant reshuffling, gene conversion, and recent transposition in eMSY.

18. Line 191. The idea of combining stratum 2 and 3 was originally introduced in 2013 by Gang Li, William Murphy et al. (Genome Research), two of the co-authors of the manuscript, not in ref. 12.

Response 18. Thank you, reference corrected.

19. Lines 237-238. “The function of these Y-born transcripts remain enigmatic, though some, like ETY7, represent novel equine non-coding RNAs and might have regulatory roles” Is there any direct evidence of this regulatory function other than testis-specific expression? Almost the entire genome is expressed in this tissue (see below).

Response 19. No, this is just speculation with no direct evidence. We argue that recent studies on the expression patterns and functions of tetrapod lncRNAs (Necsulea et al. 2014) support our speculation. On the other hand, the Reviewer has a strong argument too because, indeed, testis is a tissue which apparently allows ‘any’ sequence to be transcribed regardless of biological relevance (Soumillon et al. 2013). The truth is likely in the middle. We also would like to point out that Soumillon et al 2013 go on to hypothesize that this “promiscuous” transcription in male germ cells “may have facilitated rapid divergence of testis transcriptomes” which implies that such expression may be functionally relevant for evolution. In response to the Reviewer, we softened our suggestion but supported it with an additional reference. The text reads now as follows:

Lines 249-252: *The function, if any, of these Y-born transcripts remain enigmatic, though some, like ETY7, represent novel equine noncoding RNAs (Table S5) and may have regulatory roles in development or spermatogenesis, as recently shown for many tetrapod long noncoding RNAs*²⁹.

20. Lines 259-260. “No ETSTY7 sequences were detected in the rhinoceros”. The analyses shown are insufficient. If ETSTY7 is a long-non-coding RNA, selection acts on these elements to preserve the structure but not the nucleotide sequence, except during short-evolutionary periods. Horse and zebra diverged 4-7 MYA, so FISH analyses could still recover a signal. However, horse and rhinoceros diverged 55 MYA. Given the noncoding nature of ETSTY7, FISH does not seem to be the best approach to detect or discard the presence of this gene in distant species. This needs to be corrected.

Response 20. The reviewer is correct about the limitations of FISH in resolution and sensitivity. The fact that we produced detectable FISH signals with a 600 bp PCR product as a hybridization probe in horses, asses and zebras indicates that the sequence is extremely abundant in these species. On the other hand, we agree that the absence of FISH signal in rhinoceros is inconclusive and may indicate lack of sufficient sequence homology or copy numbers to be detected by FISH, or true absence of these sequences in rhino. Thus, we conducted additional experiments by PCR with *ETSTY7* primers using gDNA of equids, Perissodactyls (rhino, tapir) and a random selection of male mammals (dog, dolphin, yak, armadillo) from diverged eutherian

orders. We detected *ETSTY7* sequences by PCR in horses and equids only, and not in rhino or other species. In response, additional evidence to this claim is presented in **Figure 6, Supporting Information** and **Figure S8**. Manuscript text reads now as follows:

Lines: 275-276: *No ETSTY7 sequences were detected by FISH or PCR in the rhinoceros, an evolutionarily distant Perissodactyl species (SI; Figs. 6; S7; S8).*

Supplementary Information 5.2. *ETSTY7* sequence distribution in equids:

In contrast, no hybridization with ETSTY7 was detected in the rhino (Fig. S8). This may indicate that these sequences are too diverged and/or with too few copies for being detected by FISH, or that ETSTY7-homologous sequences are not present in other Perissodactyls. To test the latter, we conducted experiments by PCR with ETSTY7-3 ex 3 primers using gDNA of equids, Perissodactyls (rhino, tapir) and a random selection of mammals from diverged eutherian orders. ETSTY7 was amplified by PCR in horses and equids only, and not in rhino or other mammalian species (Fig. S8).

21. Lines 273-274. The fact that *ETSTY7* could come from the parasite *Parascaris* is an extraordinary claim, and as such, it needs extraordinary evidence. The evidence shown is not convincing enough. The authors need to show, beyond any reasonable doubt, that the libraries sequenced did not contain contaminating *Parascaris* DNA. Intestinal parasites can sometimes be found in the host bloodstream and their DNA could have therefore been mixed with the horse DNA during whole-DNA extraction. Furthermore, parasites often contain genomes with low GC content. It is therefore not unlikely that if a mix of horse/parasite DNA was sequenced, some low-GC-low-complexity sequences from the Y chromosome and the parasite were combined to form chimeric sequences. The fact that one particular region in the horse *ETSTY7* gene is similar to *Parascaris* (intron 2/3 and exon 3) is exactly what one could expect if this gene were a chimeric sequence. The authors need to show that the libraries are free of any contaminating *Parascaris* sequences, that the *ETSTY7* gene (including intron 2/3 and exon 3) is located in one single DNA molecule, and that the *ETSTY7* gene is indeed absent in other mammalian species.

Response 21. We conducted additional PCR experiments with primers derived from horse-specific, *Parascaris*-specific and horse-parasite shared sequences on DNA templates originating from multiple individual adult worms, eggs and larvae, as well as from horse/equids gDNA and on DNA of 10 horse autosomal BACs and 12 X and Y BAC clones. We do not find evidence for *ETSTY7* sequences being a result of parasite-to-horse contamination, chimerism, or horse-to-parasite contamination. Additional evidence to this claim is detailed in response to Reviewer #2 and presented in essentially revised **Figure. 6**, Supplementary Information and a newly added **Supplementary Figure S8**.

22. Lines 278-279. “A multiple sequence alignment of horse-derived and *Parascaris*-derived *ETSTY7* segments was used to reconstruct a phylogenetic tree, which was polyphyletic (Fig. 6)” Nothing can be concluded from the trees shown in Fig6. The trees are unresolved.

Response 22. We agree that the tree we showed was not very informative and so we performed a more thorough phylogenetic analysis. We added 46 more sequences including 14 from domestic horse, 15 from Przewalski’s horse, and 18 *ETSTY7*-like donkey from genome assemblies. Further, we PCR-amplified and sequenced *ETSTY7* in one male and one female horse (2 sequences), and additional *Parascaris* adults, eggs, and gonads (3 sequences, which matched the

previous sequences in our original submission). The 46 additional ETSTY7-like sequences were combined with the 27 sequences we originally analyzed. We aligned them with MUSCLE and reconstructed the phylogenetic tree using maximum likelihood with 1000 bootstrap replicates in RaxML which is more informative than what we had originally. Please also see Response 5 to Reviewer #1. Essential revisions are made in **Figure 6** and sequence data added into **Table S6**.

23. Lines 284-287. “Altogether, our data provide strong evidence of horizontal transfer (HT) between the horse and its parasite of a putatively functional ampliconic sequence. To the best of our knowledge, this is the first HT described in equids, and among only a few verified HTs within vertebrates.” The authors do not show sufficient evidence to support this claim.

Response 23. Additional evidence to this claim is detailed in response to Reviewer #2 and presented in essentially revised **Figure 6, Supplementary Information** and a newly added **Supplementary Figure S8**.

24. Lines 290-291. “The testes-specific transcription of ETSTY7 suggests it functions in male fertility in the horse.” Here the authors take a gigantic step from performing superficial analyses to major biological claims. All chromosomes (autosomes and sex chromosomes) have a spurious testis-specific transcription. Deep sequencing of the testis transcriptome of mouse resulted in around 15,000 protein-coding genes expressed in testis and countless numbers of non-coding RNAs and repeated elements. Is it safe to assume that all of these transcripts have a function in fertility and spermatogenesis? No. Testes have a massively noisy expression due to the histone/protamine replacement and the removal of methylation markers, which make almost all the genome accessible to transcription. Therefore, testis-expression does not indicate, by no means, male fertility. I encourage the authors to read the literature regarding the noisy transcription in testis due to the histone/protamine replacement.

Response 24. We agree with the reviewer that the suggestion about *ETSTY7* possible functions in male fertility is premature. Thank you for reminding about the peculiarities and relaxed regulation of testis transcription (e.g. Soumillon et al. 2013; Necșulea et al. 2014). These features of testis transcriptome certainly call for caution. Though on the other hand, we can argue that based on solid experimental data in human and mouse, several Y-linked testis transcripts are critical for spermatogenesis and male fertility. In our case, we do not have such solid data and we gladly tone down about *ETSTY7* functionality. The text reads now as follows:

Lines 312-315: *Of immediate interest is to determine the origin of ETSTY7 (e.g., transposon, noncoding RNA), and whether these sequences have any biological functions. Testis-specific expression alone provides limited clues because the permissive epigenetic regulation of testis allows transcription of many potentially nonfunctional sequences*³².

25. Lines 324-325. “We hypothesized that MSY genes with comparable expression in horse and donkey testis but significant dysregulation in mules, likely have roles in spermatogenesis.” Why would this be the case? I have some difficulty understanding the logic behind this hypothesis. Changes in gene expression in mules compared to donkey and horse may be explained due to general genetic hybrid incompatibility, unrelated to male fertility. The idea of comparing donkey, horse, and mule is fine. But the comparison needs to be performed at a genomic level. That is, how many genes (autosomal and sex-linked) are similar in horse and donkey but different in mules? How many (autosomal and sex-linked) are related to spermatogenesis? Are

the three Y genes peculiar (important) cases or are among hundreds of miss-regulated genes in mules?

Response 25. By generating testis RNAseq data for the two equids and their hybrid, we acquired data genome-wide and not just for the Y as presented in Supplementary Data File 1. However, genome-wide RNAseq comparisons between species are problematic due to sequence divergence. To account for this, the alignment parameters were relaxed for this experiment (See Methods).

We acknowledge the fact that hybrid incompatibility could lead to mis-expression of testis transcripts, and have changed the text to reflect this. In addition, we performed 3-way genome-wide fold-change comparison, identified genes with >30-fold difference between horse/donkey and the mule, and conducted GO enrichment for these genes across the genome. Genome-wide analysis supports our original assumption that genes with comparable expression in horse and donkey testis but significant dysregulation in mules have roles in male fertility. Based on the Reviewer's comments and additional data analysis, we made the following changes in the manuscript:

a) Change to language in **Candidate MSY genes for stallion fertility, page 9**:

Lines 338-340: *We hypothesized that MSY genes with comparable expression in horse and donkey testis but significant dysregulation in mules, **may** have roles in spermatogenesis **or reflect the general genetic hybrid incompatibilities.***

Lines 351-353: *Functional dysregulation of these genes in mule testis suggests their role in spermatogenesis and **is supported by genome-wide GO enrichment analysis (SI; Supplementary Data File1).***

b) Changes in Supplementary Data File 1 (SDF1): the file contains now 3 spreadsheets denoted as:

SDF1.1 – genome-wide RNAseq read counts in horse donkey and mule; we added columns for chromosomal location, coverage comparison between horse-mule and donkey-mule, and list of the loci where both horse and donkey had 30x greater read depth over the mule.

SDF1.2 – Genome-wide list of genes significantly down-regulated in mules compared to both horse and donkey.

SDF1.3 - GO Enrichment for genes with 30x greater read depth in horse and donkey compared to mule

c) Added a paragraph to Supplementary Information **4.4. MSY testis transcripts in horse, donkey and mule:**

Genome-wide analysis of 29,371 transcripts (Supplementary Data File1) showed that 839 differed by greater than 30x read depth in both donkey and horse when compared to mule. Of these, 268 have no Ensembl annotation. Of the remaining 571 transcripts, 543 were functionally mappable using gene ontology enrichment with PANTHER 13.1 (10.1093/nar/gks1118). The overrepresentation test with GO Ontology database built 2018-02-02 produced 36 over-represented biological processes at FDR < 0.05. Of these, at least 30 are directly involved in male fertility.

REVIEWERS' COMMENTS:

Reviewer #1 (Remarks to the Author):

The authors have revised the manuscript, and my comments and those of the other reviewers have been addressed. The revised figures and text are much clearer now (Figure 4 especially). I am happy to recommend the manuscript for publication.

The additional phylogenetic analysis of ETSTY7 is useful, and I agree with the authors that more work will be needed to address functionality. Indeed, there are many interesting questions to follow up on, and I look forward to the authors' future work.

Reviewer #2 (Remarks to the Author):

In the revised version of the manuscript "Horse Y chromosome assembly displays unique evolutionary features, putative stallion fertility genes and horizontal transfer" the authors performed additional experiments to address my major concern regarding the unprecedented observation of horizontal gene transfer occurred between the horse and the parasite *Parascaris* ssp.

They made additional controls, revised Figure 6 and added an additional Figure (S8) documenting their approach and the results. The additional experiments convince me to agree with their assumptions. I appreciate all the efforts done by the authors and recommend this article for publication.

one minor comment

Fig 6B: the lane description fits only for the upper two panels (ETsTY7-3 and PEQ001). In the lower three panels, less samples are loaded on the gel and lane description is not appropriate.

Reviewer #3 (Remarks to the Author):

I think the authors made an excellent work clarifying their aims, methods, references, conclusions and more importantly, the novelty of their results, specifically, the differences between the current manuscript and their publication from 2011. Overall, the new manuscript is sound and convincing. Despite the new set of evidence shown, I'm still not sure of the parasite-horse HTG. However, I'm convinced that future studies will explore this extraordinary finding in detail. I'm happy with the current version of the manuscript and I have no further comments.

Manuscript NCOMMS-17-32840A

Response to reviewers' comments

We thank the referees for their positive comments. As follows, please find our response to the additional comment by Reviewer #2 together with the description of revisions made in the manuscript.

REVIEWERS' COMMENTS:

Reviewer #1 (Remarks to the Author):

The authors have revised the manuscript, and my comments and those of the other reviewers have been addressed. The revised figures and text are much clearer now (Figure 4 especially). I am happy to recommend the manuscript for publication.

The additional phylogenetic analysis of ETSTY7 is useful, and I agree with the authors that more work will be needed to address functionality. Indeed, there are many interesting questions to follow up on, and I look forward to the authors' future work.

Reviewer #2 (Remarks to the Author):

In the revised version of the manuscript "Horse Y chromosome assembly displays unique evolutionary features, putative stallion fertility genes and horizontal transfer" the authors performed additional experiments to address my major concern regarding the unprecedented observation of horizontal gene transfer occurred between the horse and the parasite *Parascaris* ssp. They made additional controls, revised Figure 6 and added an additional Figure (S8) documenting their approach and the results. The additional experiments convince me to agree with their assumptions. I appreciate all the efforts done by the authors and recommend this article for publication.

one minor comment

Fig 6B: the lane description fits only for the upper two panels (ETSTY7-3 and PEQ001). In the lower three panels, less samples are loaded on the gel and lane description is not appropriate.

RESPONSE: We thank the reviewer for noticing this error, which is due to the multiple PCR experiments conducted for HT validation, where we gradually increased the number of test templates as DNA was isolated from additional *Parascaris* samples. We corrected **Figure 6b** by replacing panels 3, 4 and 5. Now all 5 panels have the same 16 DNA templates.

Reviewer #3 (Remarks to the Author):

I think the authors made an excellent work clarifying their aims, methods, references, conclusions and more importantly, the novelty of their results, specifically, the differences between the current manuscript and their publication from 2011. Overall, the new manuscript is sound and convincing. Despite the new set of evidence shown, I'm still not sure of the parasite-horse HTG. However, I'm convinced that future studies will explore this extraordinary finding in detail. I'm happy with the current version of the manuscript and I have no further comments.